# Deformation- and damage-free transfer of soft electronics onto highly curved and fragile biological surfaces

Kyeong Min Song [1,2,9], Myung-Kun Chung [3,9], Jeehoon Jung [4], Jungjae Park[5], Min-Uk Kim [3], Jae-Young Yoo [6], Minsu Park[7], Geumbee Lee[8], Jeehyun Hong[2], Jun-Bo Yoon [3] ✉ & Yeon Sik Jung [2] ✉

Seamless integration of soft electronics with biological tissues enables high-fidelity physiological sensing by maintaining intimate mechanical and thermal contact. However, transferring these devices onto highly curved and fragile surfaces remains challenging, as conventional methods often induce strain, thermal distortion, or tissue damage. To address these challenges, we introduce a deformation-decoupling, adhesion-switchable yield-stress fluid (DAYS-fluid) that enables non-invasive, damage-free transfer of soft electronics onto complex biological surfaces, including those with undercuts and negative curvature. DAYS-fluid undergoes a reversible solid-to-liquid transition at an ultra-low yield stress of 0.0253 kPa, far below the rupture pressure (~0.2 kPa) of extremely fragile biological substrates such as raw egg yolk. This low yield stress, combined with tunable viscosity, decouples fluid motion from embedded electronics, preserving structural integrity during transfer. A water-triggered adhesion-switching mechanism further reduces interfacial adhesion to near zero, enabling gentle detachment. We demonstrate high-fidelity thermal sensing on moving joints, without compromising device performance or tissue integrity.

Recent progress in materials engineering and mechanical design has enabled semiconductor microfabricated bio-interfaced electronics to achieve stable signal acquisition and multimodal sensing, while maintaining intimate mechanical coupling with soft tissues. These advances now allow microfabrication-level precision to be applied directly on the skin for noninvasive, real-time physiological monitoring[1–3]. Nevertheless, most device fabrication still depends on rigid, planar wafer substrates, making an additional integration step essential when interfacing with curved or mechanically fragile biological surfaces. Among existing integration strategies, transfer printing has emerged as the most versatile approach. It bridges planar semiconductor microfabrication with three-dimensional biological targets by enabling precise and deterministic placement of ultrathin electronic structures onto nonplanar surfaces. Other routes, including in situ patterning,

[1]Energy and Environmental Division, Korea Institute of Ceramic Engineering and Technology, 101, Soho-ro, Jinju-si, Gyeongsangnam-do, Republic of Korea. [2]Department of Materials Science and Engineering, Korea Advanced Institute of Science and Technology (KAIST), 291 Daehak-ro, Yuseong-gu, Daejeon, Republic of Korea. [3]School of Electrical Engineering, Korea Advanced Institute of Science and Technology (KAIST), 291 Daehak-ro, Yuseong-gu, Daejeon, Republic of Korea. [4]Low Carbon Energy R&D Center, Korea Institute of Industrial Technology, 55, Jongga-ro, Jung-gu, Ulsan, Republic of Korea. [5]Research Institute of Industrial Science and Technology, 156, Jungheung-ro, Buk-gu, Pohang-si, Gyeongsangbuk-do, Republic of Korea. [6]Department of Semiconductor Convergence Engineering, Sungkyunkwan University (SKKU), 2066, Seobu-ro, Jangan-gu, Suwon-si, Gyeonggi-do, Republic of Korea. [7]Department of Polymer Science and Engineering, Dankook University, 152, Jukjeon-ro, Suji-gu, Yongin-si, Gyeonggi-do, Republic of Korea. [8]Department of Chemical Engineering, Kyungpook National University, 80, Daehak-ro, Buk-gu, Daegu, National University, Daegu, Republic of Korea. [9]These authors contributed equally: Kyeong Min Song, Myung-Kun Chung. ✉e-mail: jbyoon@kaist.ac.kr; ysjung@kaist.ac.kr

direct printing, and integration on thick stretchable substrates, have been explored, but they cannot directly accommodate micro-fabricated semiconductor devices, thereby limiting device complexity, resolution, and scalability (Supplementary Note 1, Supplementary Table 1)[4,5].

This need for an effective integration strategy becomes particularly pronounced when targeting convex, dynamically contoured biological surfaces where clinically relevant physiological signals are generated. Such geometries including finger and knee joints, retinal curvature, and various internal organs are mechanically active and highly three-dimensional, making them especially challenging for conformal electronic integration (Supplementary Fig. 1). Convex joints are especially prone to inflammation and structural degradation under repetitive mechanical loading, often resulting in localized changes in thermal and mechanical patterns that serve as early indicators of tissue damage[6,7]. Similarly, highly curved anatomical regions such as the cardiac apex and the superior surface of the kidney are critical sites for electrical signal mapping and hemodynamic assessment. These locations provide sensitive biomechanical cues for diagnosing tissue dysfunction and detecting acute transplant rejection[8,9]. Integrating bio-interfaced electronics with such geometrically complex biological substrates holds significant potential for advancing physiological monitoring, enabling early-stage disease detection, and informing precision therapeutic strategies.

Nevertheless, even state-of-the-art transfer printing techniques remain limited in their ability to reliably integrate electronics onto highly curved biological surfaces. A central challenge lies in the lack of suitable transfer carriers. During contact or release, existing media often impose mechanical or thermal stresses that can damage both the electronic devices and the underlying biological tissues (Supplementary Table 2). Traditional solid carriers, such as elastomers[10–12] and adhesive tape-based systems[13], often deform electronic structures when applied to complex geometries, resulting in misalignment, poor adhesion, and compromised performance. Moreover, the mechanical stress required for conformal contact can destabilize both the electronics and the underlying biological tissues. Alternative carriers such as water bath–based hydro transfer and thermally activated materials including molten sugar[14–17] offer improved conformability but involve complex handling procedures, limiting their practicality for bio-interfaced applications. can offer improved conformability. However, these approaches require complex handling procedures, which limit their practicality for bio-interfaced applications. In particular, molten sugar requires thermal heating above 60 °C, and its intrinsically high liquid-state viscosity generates significant mechanical stress during transfer, which can damage fragile electronics and soft biological substrates. Such thermal cycling and the associated solid–liquid phase transitions render it unsuitable for soft or temperature-sensitive biological substrates as well as fragile electronic devices (Supplementary Table 3 and 4). These limitations underscore the need for a transfer medium capable of accommodating complex topographies while preserving both device integrity and biological compatibility—an essential step toward the seamless integration of electronics with fragile, anatomically challenging surfaces.

In this study, we introduce a deformation-decoupling, adhesion-switchable yield-stress fluid (DAYS-fluid) that enables efficient, non-invasive transfer of soft electronics onto highly curved and mechanically fragile biological surfaces, while preserving both device functionality and tissue integrity. Composed of water and biocompatible fumed silica, DAYS-fluid undergoes a reversible transition from a solid-like to a viscous state at an ultra-low yield stress of 0.0253 kPa. This stress-responsive rheology allows the fluid to form seamless, conformal contact with complex topographies while maintaining precise control over embedded electronic structures. By tuning viscosity, fluid deformation is effectively decoupled from the electronics, preventing mechanical distortion and preserving microscale features. In parallel, a water-triggered adhesion-switching mechanism reduces surface adhesion to near zero, enabling clean, gentle detachment. This dual functionality renders DAYS-fluid well-suited for interfacing soft electronics with delicate, high-curvature surfaces, including finger joints and even raw egg yolks. As a demonstration, we achieve stable transfer of a temperature sensor array onto a moving finger joint, enabling real-time thermal monitoring during dynamic tasks such as typing and climbing.

## Results

### Mechanistic basis of DAYS-fluid for adaptive electronic transfer

DAYS-fluid operates through three integrated mechanisms (Fig. 1): (1) stress-responsive rheology, (2) deformation decoupling, and (3) switchable interfacial adhesion. These features enable DAYS-fluid to function as an adaptive transfer medium, offering mechanical stability during handling, conformal adaptability to curved surfaces, and high transfer efficiency, all while maintaining a gentle, non-destructive interface. As a result, DAYS-fluid enables non-invasive transfer of soft electronics while preserving the structural integrity of both devices and underlying biological substrates, making it well-suited for high-curvature and mechanically sensitive environments.

The core principle of DAYS-fluid derives from its stress-responsive rheological properties, which allow precise and adaptive control over the transfer and integration of electronic components (Fig. 1a–c). As a yield-stress fluid, DAYS-fluid undergoes a controlled and reversible transition between solid-like and liquid-like states when mechanical stress exceeds a yield stress threshold (Fig. 1b, c). This transition is achieved through a biphasic system consisting of water and biocompatible fumed silica nanoparticles[18–21]. These nanoparticles impart yield-stress characteristics, which are tunable within the moderate range of $10–10^3$ Pa, ensuring optimal adaptability across various application environments.

As illustrated in Fig. 1b–d, in the absence of external stress, the fumed silica particles form a percolated network[18–21], imparting solid-like behavior, which securely immobilizes the electronics. In contrast to free-flowing liquids that generate uncontrolled currents misaligning delicate electronics[7,8], this solid-like state facilitates precise placement of electronic devices onto the targeted skin location, before transfer. Upon application of stress, the silica particles align along the stress direction, inducing a reversible transition to a viscous liquid phase (Supplementary Note 2 and Supplementary Fig. 2)[18–21]. This stress-triggered fluidization allows seamless adaptation to a wide range of surface curvatures, making it particularly suitable for the integration of soft electronics with anatomically complex or highly contoured biological substrates.

The rheological behavior of DAYS-fluid was systematically optimized by tuning the concentration of fumed silica (Supplementary Fig. 3-4). An optimal yield stress range of 0.024–0.836 kPa was identified at silica concentrations between 0.11 and 0.22 wt.%. At lower concentrations (e.g., 0.06 wt.%), the fluid exhibited uncontrolled spreading, whereas higher concentrations (≥0.27 wt.%) led to nanoparticle agglomeration, reducing conformability on curved surfaces and degrading fluid performance. Given the critical importance of viscosity control, further detailed in Fig. 2, a fluid formulation with a yield stress of 0.0253 ( ± 0.00135) kPa (Supplementary Fig. 5), achieved at a silica concentration of 0.11 wt.%, was selected for experimental validation. This non-invasive capability is further demonstrated by its successful transfer onto egg yolks, which have an exceptionally low rupture pressure (0.2–0.5 kPa), far below the skin discomfort threshold (180-200 kPa) (Fig. 1f and Supplementary Fig. 6)[22–27]. These values illustrate that DAYS-fluid imposes negligible mechanical load on biological surfaces. In contrast, heat-based carriers such as molten sugar require melting and recrystallization at elevated temperatures (>60 °C), which can generate thermal surface damage and high-viscosity–induced mechanical stress during operation. DAYS-fluid, by

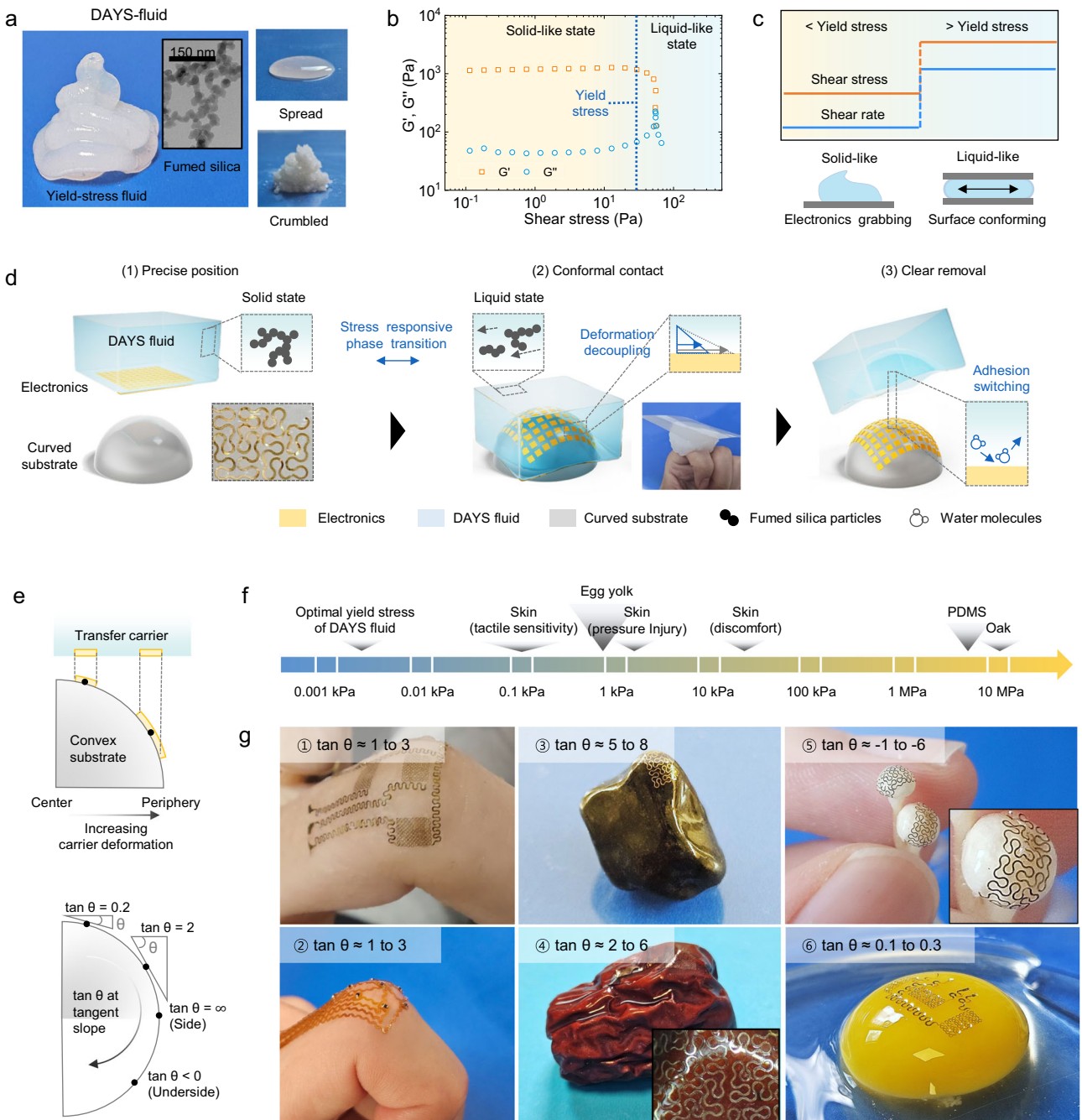

**Fig. 1 | Design features and operating principle of DAYS fluid. a** A photograph of DAYS-fluid. The inset shows a TEM image of fumed silica particles(Independently repeated at least ten times with similar results). Optical images to the right depict the fluid with low (top) and high (bottom) particle concentrations. **b** Rheological properties of DAYS-fluid: storage modulus (G') and loss modulus (G") as a function of shear stress. The intersection of G' and G" indicates the phase transition from a solid-like to a liquid-like state. **c** Concept utilizing stress responsive phase transition of yield-stress fluid for transfer printing. The shear rate changes as the shear stress increases more than the yield stress. **d** Schematics of the transfer procedure using DAYS-fluid. Inset images illustrate the key mechanisms in the printing process: phase transition, deformation decoupling, and adhesion switching. **e** Quantitative geometric analysis of convex surfaces, with the tangent angle (tan θ) calculated based on the local slope of the surface. **f** Comparative plot summarizing the compressive pressures of yield stress of DAYS fluid, skin tactile sensitivity threshold, rupture pressure of egg yolk, skin pressure threshold for pressure injury (≥1 h exposure), skin pressure discomfort threshold, failure pressure of PDMS (Sylgard 184, 10:1), and perpendicular-to-grain failure pressure of oak hardwood. **g** Optical images of electronics transferred onto diverse substrates using DAYS-fluid: ① finger joint1, ② finger joint 2, ③ sharp-edged rock, ④ dried jujube date, ⑤ mushroom cap, and ⑥ quail egg yolk. Source data are provided as a Source Data.

operating entirely under ambient conditions with an ultra-low yield stress, avoids these limitations and enables gentle, non-invasive transfer printing (Supplementary Tables 3 and 4).

Another critical advantage of DAYS-fluid is its ability to isolate fluid deformation from the electronics by minimizing viscosity, thereby enabling the seamless transfer of electronics without distortion (Figs. 1d and 2). In contrast, conventional high-viscosity fluids and conventional solid transfer carriers often introduce drag and mechanical strain on the electronic components as they transition from planar to three-dimensional geometries, compromising both

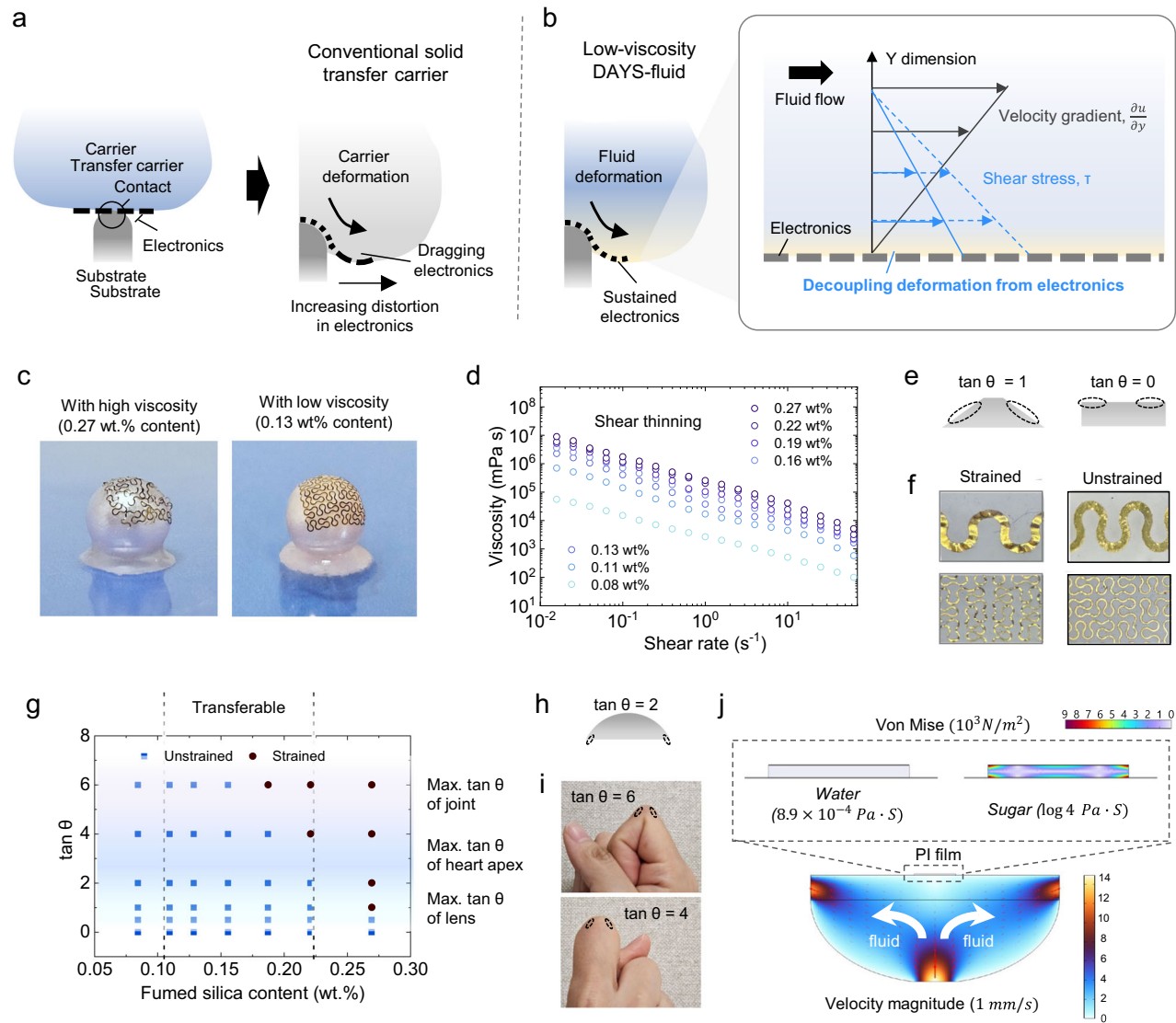

**Fig. 2 | Deformation Decoupling for Seamless Electronics Transfer on Curved Surfaces. a, b** Schematic comparing conventional solid transfer carriers and DAYS-fluid when applied to a highly curved surface. The inset illustrates the deformation decoupling mechanism, where low viscosity allows fluid deformation to occur independently of electronics. **c** Photographs of electrode arrays transferred using fluids with high viscosity (left) and low viscosity (right), highlighting the impact on transfer quality. **d** Viscosity profiles as a function of shear rate for fluids with varying fumed silica particle concentrations. **e** Schematic of a conical surface with slopes of 0 and 1, simulating different curvatures for testing. **f** Photographs of electronics showing strained (left) and unstrained (right) conditions after transfer. **g** Strain maps illustrating the relationship between substrate slope and particle concentrations. Blue and black regions represent the maximum slopes observed for joints and lenses, respectively. **h** Schematic of a contact lens with maximum tangent angle. **i** Photographs of a finger joint with maximum tangent angle. The black lines indicate the positions of ligaments and tendons. **j** Finite element analysis of flow-induced stress beneath a polyimide (PI) film (water viscosity: $8.9 \times 10^{-4}$ Pa·s; sugar viscosity: 0.6 Pa·s). This computational analysis provides further insight into the stress distribution during transfer. Source data are provided as a Source Data file.

performance and adhesion. Moreover, DAYS-fluid further enhances the transfer process through a water-triggered adhesion-switching mechanism (Figs. 1d and 3). Unlike conventional adhesives or solid transfer carriers, which require mechanical peeling or chemical treatments that can potentially damage delicate electronics and substrate, this adhesion-switching mechanical can significantly reduce the fluid's interfacial adhesion force to nearly zero, allowing soft and non-destructive detachment of the transfer medium.

With viscosity-controlled, stress-responsive flow and an efficient adhesion-switching mechanism, DAYS-fluid enables precise device transfer onto complex and previously inaccessible surface morphologies. (Fig. 1g). To systematically analyze the substrate morphology, the tangent angle (tan θ) of the local surface tangent line is used as a

quantitative parameter (Fig. 1e), This enables precise characterization of curvature variations and local slope transitions that directly influence the mechanical conformity and adhesion behavior of the transferred devices. On convex surfaces, tan θ increases toward the outermost edge, posing greater challenges for the conformal transfer of electronics. While previous studies primarily focused on low tan θ (<2) surfaces[10,11], such as lens-shaped geometries, our findings demonstrate that DAYS-fluid extends beyond these conventional limits and successfully achieves conformal transfer even on surfaces exhibiting extreme curvatures and negative slopes, conditions previously unexplored in transfer printing. Specifically, DAYS-fluid demonstrates distortion-free transfer across a broad range of geometries with distinct curvature profiles: two finger joints with

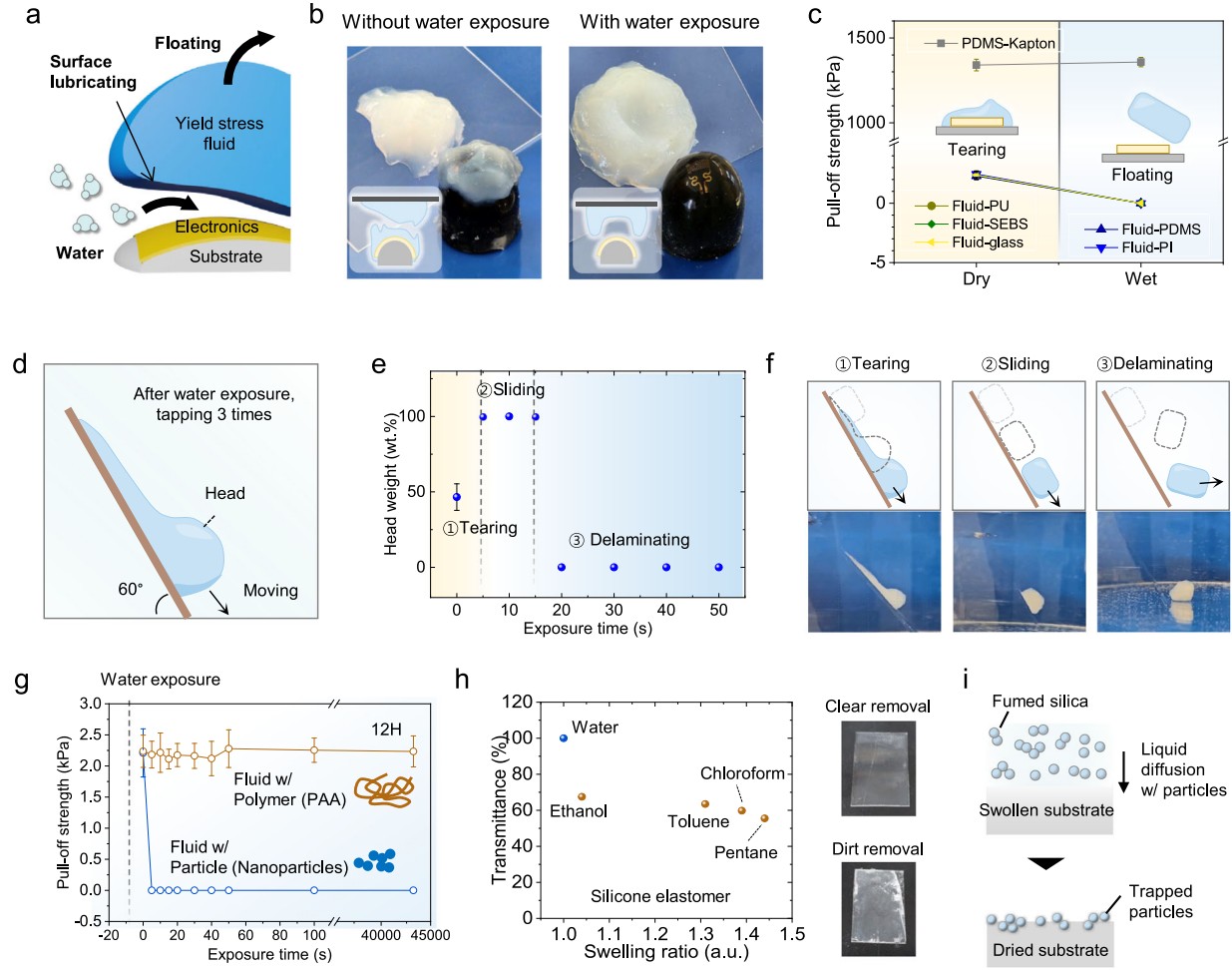

**Fig. 3 | Water-Assisted Adhesion Control for Clean Fluid Removal. a** Schematic illustration of the water-assisted adhesion-switching mechanism at the fluid surface. **b** Schematic and corresponding photograph of the fluid with and without water exposure. **c** Variation in pull-off strength under dry and wet conditions across three different interfaces: PDMS-kapton, fluid-PDMS, fluid-Kapton(PI), fluid-polyurethane(PU), fluid-SEBS, and fluid-glass (mean ± s.d., $n = 4$ independent experiments). **d–f** Fluid droplet behavior under varying water exposure times:(**d**) Schematic representation of the droplet movement, highlighting the positions of the moving head fluid. **e** Weight percentage of the moving head fluid as a function of water exposure time (mean ± s.d., $n = 4$ independent experiments). **f** Schematics and images depicting fluid droplet states: ① Tearing, ② Sliding, and ③ Delamination. **g** Pull-off strength of the fluid measured at different water exposure times. Nanoparticles and the polymer PAA were tested as representative materials (mean ± s.d., $n = 4$ independent experiments). **h, i** Significance of liquid selection in fluid composition: (**h**) Transmittance and swelling ratio of the substrate depending on the type of liquid in the fluid. Representative images of clear and contaminated substrates are shown on the right. (**i**) Schematic illustrating the process of liquid diffusion into the substrate, leading to opacity. Source data are provided as a Source Data file.

moderate convex curvature ($\tan \theta \approx 1$ to 3; Fig. 1g-①, ②), a sharp-edged rock exhibiting high convexity and discontinuous slopes ($\tan \theta \approx 5$ to 8; Fig. 1g-③), a dried fruit featuring irregular multi-curved wrinkles ($\tan \theta \approx 2$ to 6; Fig. 1g-④), a mushroom cap with negative Gaussian curvature and undercut geometry ($\tan \theta \approx -1$ to $-6$; Fig. 1g-⑤), and a soft, hemispherical quail egg yolk with smooth curvature ($\tan \theta \approx 0.1$ to 0.3 Fig. 1g-⑥). These results highlight that DAYS-fluid enables conformal, damage-free device transfer on curvature regimes that were previously unattainable with conventional carriers, demonstrating its versatility across both rigid and deformable substrates.

More notably, DAYS-fluid achieves these complex transfers without requiring heat or significant pressure, making it particularly suitable for application on fragile surfaces, while effectively preventing mechanical and thermal damage (①, ②, ④, ⑤, and ⑥ Fig. 1g). This noninvasive capability is further demonstrated by its successful transfer onto egg yolks, which have an exceptionally low rupture pressure (0.2–0.5 kPa), far below the skin discomfort threshold (180-200 kPa) (Fig. 1f and Supplementary Fig. 6)[22–27]. To further substantiate the

interfacial conformability of DAYS-fluid, we compared multiple transfer media on a mushroom substrate with negative Gaussian curvature and undercut geometry. Magnified optical and confocal images (Fig. 1g-⑤, Supplementary Note 3, Supplementary Table 5, and Supplementary Fig. 7) reveal that conventional carriers cause interfacial distortion, folding, or partial delamination, leading to non-uniform contact and strain accumulation. In contrast, DAYS-fluid achieves uniform, strain-free conformal contact across the entire interface, including undercut regions.

## Deformation decoupling for seamless electronics transfer on curved surfaces

By precisely minimizing the viscosity of DAYS-fluid, we achieved effective decoupling of carrier deformation from the electronics, thereby mitigating mechanical constraints typically encountered during the transfer process (Fig. 2a–c). This rheological optimization is particularly effective in accommodating extensive shape deformations of the transfer medium at high tangent angles ($\tan \theta$) along the

periphery of convex structures. Throughout the transfer process, the structural integrity of electronics is preserved without distortion (Fig. 2c). Viscosity can be controlled by modulating the concentration of silica particles, and a shear-thinning behavior is observed (Fig. 2d). In contrast, conventional transfer carriers induce significant mechanical stress on electronics, leading to structural misalignment and performance degradation (Fig. 2c)[10–13].

The underlying mechanism of deformation decoupling in DAYS-fluid can be effectively described by a power-law model, commonly known as the Ostwald–de Waele relationship, which characterizes the shear-thinning behavior of non-Newtonian fluids. As illustrated in the inset of Fig. 2b, ultrathin electronic films positioned on a flowing fluid surface experience a nearly uniform shear stress distribution across their thickness. According to the power-law model, this shear stress can be mathematically expressed as follows[28]:

$$\tau = K \cdot \left(\frac{\partial u}{\partial y}\right)^n \qquad (1)$$

where $\tau$ represents the shear stress, $K$ is the flow consistency index (Pa·s$^n$), $\partial u / \partial y$ denotes the shear rate or velocity gradient perpendicular to the plane of shear (s$^{-1}$), and $n$ is the flow behavior index (dimensionless). The relationship between shear stress and viscosity is given by ref. 28:

$$\eta = K \cdot \left(\frac{\partial u}{\partial y}\right)^{n-1} \qquad (2)$$

where $\eta$ is an apparent or effective viscosity (Pa s). As these expressions show, viscosity directly governs the shear stress within the fluid. Lowering the viscosity reduces the shear stress transmitted to the electronic components, thereby minimizing mechanical strain during transfer. Through careful viscosity tuning, DAYS-fluid enables the deformation of the transfer medium to occur independently of the embedded electronics, effectively preventing mechanical interference and ensuring precise device alignment.

Our approach is further validated through comprehensive and systematic investigations conducted on truncated conical surfaces with fixed tangent slopes (Fig. 2e–i), providing empirical support for the deformation-decoupling mechanism described by the power-law model. Figure 2g presents a strain distortion map of transferred electronics as a function of fluid particle concentration and the tangent angle of the substrate. We classified electronic devices with strain exceeding 5% as strained, and those below 5% as unstrained (Fig. 2f). Representative biological surfaces—convex joints, the cardiac apex, and soft contact lenses, are highlighted in blue, green, and gray, respectively. As illustrated in Fig. 2g, decreasing the silica particle concentration, and thus reducing fluid viscosity, results in lower strain levels within the electronics. These findings unequivocally substantiate the role of viscosity-controlled rheology in mitigating mechanical stress and preserving structural integrity during the transfer process.

To quantitatively assess the efficacy of our deformation-decoupling strategy, we conducted finite element method (FEM) simulations to examine the stress distribution on the electronic substrate during the transfer process. As shown in Fig. 2j, the fluid is introduced through a single inlet and exits through two outlets beneath a PI film. When water (viscosity = $8.9 \times 10^{-4}$ Pa·s) serves as the fluid medium, the stress transferred to the PI film remains minimal, indicating efficient decoupling between fluid flow and the electronics. In contrast, replacing water with a highly viscous sugar-based fluid (viscosity = 0.6 Pa·s) substantially elevates the stress exerted on the PI film.

## Gentle and non-destructive carrier detachment enabled by water-assisted adhesion switching

Unlike conventional transfer carriers that necessitate mechanical peeling or aggressive chemical treatments, the DAYS-fluid employs an adhesion switching technology that offers clean, efficient, and delicate detachment of the DAYS-fluid while preserving both device integrity and the underlying biological substrate (Fig. 3a). This process involves applying water over the adhered fluid or immersing it in a water bath, allowing water molecules to access the interface between the fluid and electronics. As these molecules diffuse, they effectively weaken adhesion forces, leading to spontaneous and soft detachment of the fluid while leaving the electronic components securely adhered to the receiving substrate (Fig. 3b). Without the use of water, however, the fluid tends to tear and leave behind remnants. This unique water-assisted adhesion-switch mechanism, which operates at the fluid's particle-rich surface, sets our method apart from traditional solvent-based techniques that require plasticizing polymer films[29,30].

A key advantage of this water-assisted mechanism is the ability to reduce the adhesion force to nearly zero, independent of substrate type. As shown in Fig. 3c (Supplementary Note 4), we measured the pull-off strength of the fluid on materials commonly used as adhesive and protective layers in soft electronic systems. In the absence of water, the fluid demonstrates significant pull-off strength, resisting detachment from the substrate. However, upon the introduction of water, the fluid detaches effortlessly, reducing pull-off strength to nearly zero. This substantial reduction in adhesion enables reliable and non-destructive transfer printing of electronics onto a variety of surfaces, greatly enhancing the practicality and efficiency of our system.

To further investigate this adhesion switching mechanism, we conducted detailed studies on the behavior of fluid droplets under varying water exposure times (Figs. 3d–f). Fluid droplets were initially placed on a slide glass and exposed to water for controlled intervals. Their mobility was then assessed by tilting the slide to 60° and applying three mechanical taps (Fig. 3d). As depicted in Fig. 3e, where the left y-axis indicates the weight of the moving droplet head, three key observations emerged: (1) Before water exposure, the droplet tore apart upon tapping, leaving behind a noticeable residue. This indicates strong adhesion to the surface. (2) After 5 seconds of water exposure, the droplet slid smoothly across the surface, leaving no residue. In this intermediate sliding regime, water molecules are hypothesized to diffuse into the interface between the fluid and the electronics, as well as the fluid and the substrate, effectively lubricating the contact surface and significantly reducing adhesion. (3) With more than 20 seconds of exposure, the fluid underwent complete detachment, freely rolling over the surface (Fig. 3f). In addition, the adhesion-switching kinetics were also found to depend on environmental humidity and temperature, where enhanced interfacial water diffusion under humid or warm conditions accelerated detachment, whereas dry or low-temperature environments delayed it (Supplementary Note 5 and Supplementary Fig. 8). This signifies a near-total loss of interfacial adhesion, allowing the fluid to completely separate from the surface.

To further validate this hypothesis, an additional experiment was conducted with fluid droplets placed on a pre-wetted slide glass (Supplementary Fig. 9). In this scenario, the fluid droplet exhibited similar sliding behavior to those exposed to water for 5 seconds. Finally, with prolonged exposure (greater than 15 seconds) on the pre-wetted surface, the droplet rolled off, further reinforcing the critical role of interfacial water in adhesion reduction.

We hypothesize that this adhesion switching behavior arises from the strong hydrogen bonding between the high–surface-area fumed silica network and interfacial water molecules, which confines hydration to the outermost region of the fluid (Supplementary Note 2). The silica–water interaction forms a dense hydrogen-bonded network that resists bulk penetration of water, thereby localizing the diffusion pathway along the interface. This interfacial hydration weakens the

yield-stress-supported mechanical interlocking at the contact boundary, leading to a rapid and reversible reduction in adhesion. Supporting this interpretation (Supplementary Note 6 and Supplementary Fig. 10), when the intrinsic water content of the fluid was intentionally increased, the overall hydrogen-bonding force between silica particles decreased, leading to weakened structural cohesion. Under these conditions, externally applied water readily diffused into the bulk rather than remaining interfacially confined, resulting in fluid dispersion instead of controlled interfacial detachment. Conversely, under optimal hydration, interfacial diffusion was limited to the surface, enabling clean, residue-free adhesion switching.

Importantly, as shown in Fig. 3c, this adhesion-switching behavior remains consistent across substrates with nearly threefold variation in surface energy[31], including polydimethylsiloxane (PDMS, ~20–24 mJ $m^{-2}$), polyimide (Kapton, PI, ~40–47 mJ $m^{-2}$), polyurethan (PU, ~35–45 mJ $m^{-2}$), styrene–ethylene–butylene–styrene (block copolymer) (SEBS, ~22–25 mJ $m^{-2}$), and glass (~65–75 mJ $m^{-2}$). This substrate-independence arises because adhesion in DAYS-fluid is governed not by wetting or surface-energy–dependent interactions, but by yield-stress–mediated mechanical gripping: the fluid plastically deforms into nanoscale asperities and forms mechanically interlocked contacts supported by its percolated silica network (Supplementary Note 4).

The adhesion-switching behavior of yield-stress fluids is highly dependent on the nature of the dispersed phase, particularly its interaction with water at the interface. To investigate the role of fluid constituents in water-assisted adhesion switching, we compared yield-stress fluids formulated with silica nanoparticles and hydrophilic polymers. As shown in Fig. 3g, the pull-off strength of yield-stress fluids composed of silica nanoparticles and hydrophilic polymers (poly-acrylic acid, PAA) is presented, with composition of 0.15 and 0.2 wt.% respectively. The nanoparticle-based formulation exhibited rapid adhesion switching after brief water exposure, consistent with the behavior observed in DAYS-fluid. In contrast, the PAA-based fluid showed no appreciable reduction in adhesion even after 12 h of water exposure. This disparity likely arises from the lower water diffusion rate in polymer matrices and stronger adhesive interactions under wet conditions[30,32,33].

In addition to fluid formulation, the proper selection of adhesion-switching solvent—that is, the solvent applied to trigger interfacial debonding, plays a critical role in achieving residue-free removal of the yield-stress fluid. As shown in Fig. 3h, we evaluated the swelling ratio and transmittance of the receiver substrate (polydimethylsiloxane, PDMS) with different solvents exposure (water, ethanol, toluene, chloroform, and pentane). The swelling ratio is based on values reported in the previous literature[34]. The transmittance was measured after removal of the yield stress fluid exposed to the solvent for 30 s. Decrease in transmittance indicates remaining residues. All tested fluids contain fumed silica particles at the same concentration of 0.11 wt.%. The substrate (PDMS film) exhibits decreased transmittance when the solvent swells the material, indicating that solvent penetration disrupts the surface structure and retains particle residues (Fig. 3i). These results indicate the importance of selecting proper solvents that do not permeate or chemically interact with the target substrate in order to ensure clean removal.

## Real-Time sensing demonstrations on highly curved surfaces using DAYS-fluid transfer

To validate the applicability of DAYS-fluid as a transfer medium for conformal integration of electronics onto convex, anatomically contoured biological surfaces, we implemented a thermal sensor array on the proximal interphalangeal joint of the index finger and conducted real-time temperature monitoring during motion (Fig. 4a). The skin of the finger joint was chosen as a representative soft biological surface because it combines pronounced curvature with mechanical softness, making it one of the most challenging regions for conformal device integration. Indeed, conformal electronic mapping on this narrow anatomical site has rarely been demonstrated. A finger, mounted thermal sensor platform, enables continuous assessment of localized temperature variations, which are clinically relevant for the early detection of overuse injuries. Such injuries are often characterized by repetitive mechanical loading that induces skin temperature changes associated with underlying inflammatory and vascular responses[35,36]. While infrared (IR) thermography is widely used for skin temperature mapping, its effectiveness for dynamic monitoring and localized analysis is limited by motion artifacts and environmental interference[37,38].

By leveraging the stress-responsive rheological properties, deformation decoupling, and adhesion-switching mechanisms of DAYS-fluid, we achieved distortion-free, high-fidelity transfer of the temperature sensor array (Fig. 4a). The thermistors, encapsulated in PI, were strategically placed: sensor channel numbers (Ch. No.) of 2, 3, 6, and 7 over the inflammation-prone collateral ligaments, and Ch. No. 5 over the fracture-prone proximal phalanx[39] (Fig. 4b). This perfect conformability ensures accurate and reliable temperature measurements during dynamic movements, which is crucial for applications that demand high precision, such as medical diagnostics and performance monitoring.

Figure 4c presents an exploded-view illustration of our wireless thermal sensor system, which is composed of two interconnected flexible printed circuit boards (fPCBs). The primary fPCB incorporates a thermistor array arranged in a serpentine layout, designed to conform seamlessly to the finger's curved surface while accommodating natural joint movement. The second fPCB incorporates a Bluetooth low-energy (BLE) system-on-chip (SoC) and related circuitry, including a voltage divider for thermistor resistance measurements. The two fPCBs connect via thin electrical wire. Notably, the BLE fPCB features a compact 4 cm × 4 cm footprint, making it well-suited for various applications such as sports, rehabilitation, and daily tasks. Further details of the wireless thermal sensor system are described in the Methods section and the Supplementary Fig. 11.

To validate the performance of the transferred device, we compared temperature sensing accuracy with and without conformal adhesion over five independent trials (Fig. 4d). The DAYS-fluid enabled seamless adhesion, resulting in accurate and consistent temperature readings during movement, with a low standard deviation of 7.27 Ω. In contrast, devices transferred using high-viscosity fluids exhibited poor adhesion, causing some channels to dangle and substantially degrading data reliability, as reflected by a much larger variation of 416.64 Ω. Additionally, these dangling devices often became entangled, contributing to user discomfort. Furthermore, as shown in Fig. 4e-g and Supplementary Fig. 12, the thermal sensor array transferred by DAYS-fluid provided stable skin temperature readings, independent of posture, angle, or distance. In contrast, IR imaging showed errors exceeding 1.01 °C due to posture-dependent detection areas and failed to measure accurate skin temperature at greater distances where the readings increasingly reflected ambient temperature. The calibration curve for resistance versus temperature is presented in Supplementary Fig. 13.

Our real-world tests further highlight the effectiveness of DAYS-fluid. In the first experiment, participant was equipped with a thermal sensing array mounted on the left index finger joint while performing a typing task (Fig. 4h–k). Repetitive tasks such as typing or writing can often lead to repetitive strain injury (RSI), causing inflammation of the ligaments and tendons[40]. As shown in Fig. 4j, k and Supplementary Fig. 14, the conformally transferred electronics provided continuous and low-artifact temperature signals on a moving finger, maintaining stable readings even during rapid finger motion. In contrast, sensor channels that failed to achieve intimate contact, as well as temperatures measured using IR imaging, exhibited continuous fluctuations and unstable behavior.

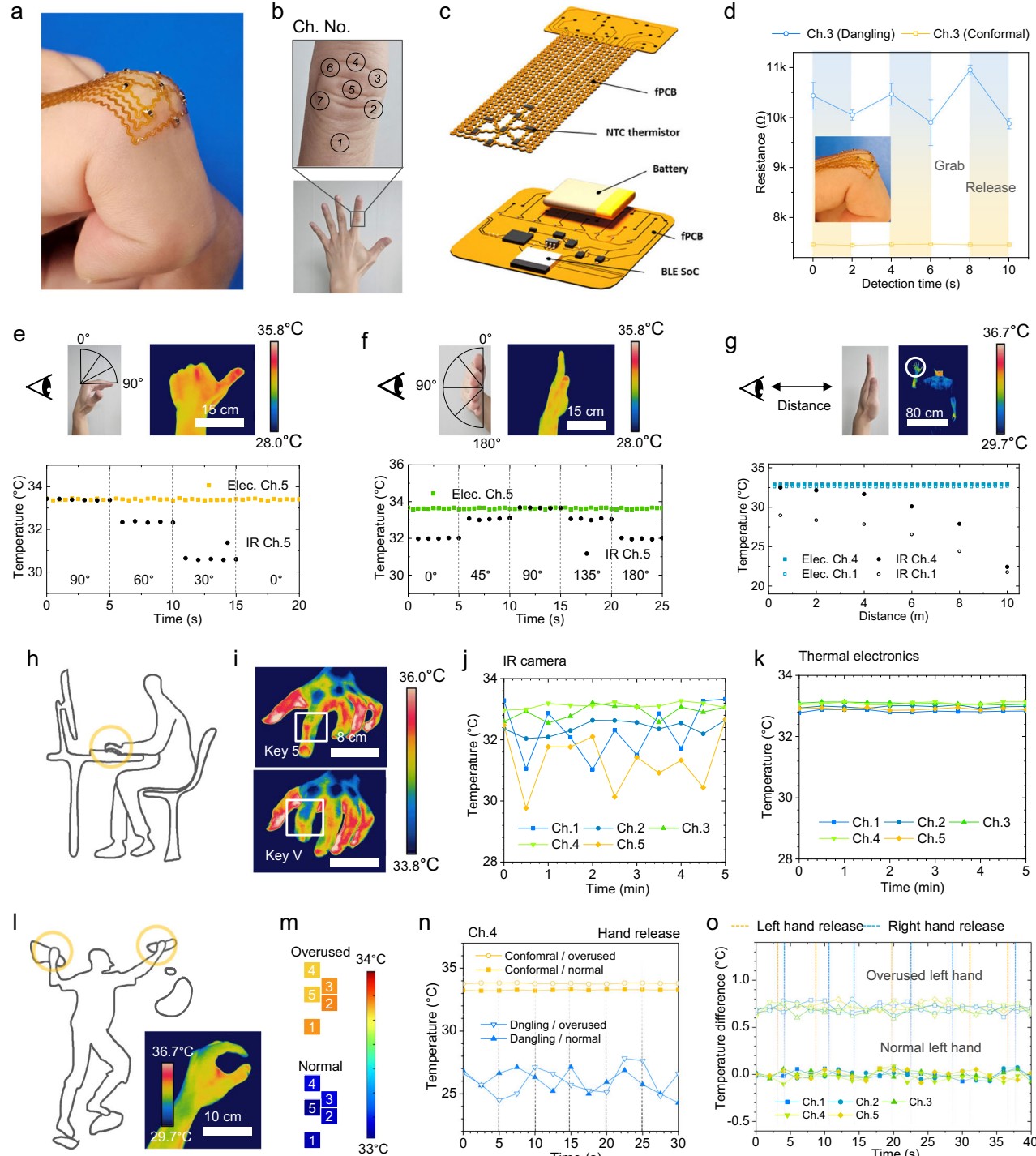

**Fig. 4 | Real-time temperature monitoring on highly curved skin surfaces.**
**a** Photograph of thermal arrays on the index finger. **b** Photographs of index finger joint indicating the positions of the channels. **c** Schematic illustration of the device layout. **d** Real-time resistance changes of electronics with and without conformal transfer. Inset image shows the electronics without conformal transfer (mean ± s.d., $n = 5$ independent experiments). **e**–**g** Real-time thermal measurements of the finger joint under varying hand postures (**e**, **f**) and distances from the camera (**g**). The upper left images display the experimental setup, while the upper right images show IR camera captures of the hand. The bottom graphs illustrate temperature changes recorded by the thermal sensor array and the IR camera. **h**–**k** Real-time thermal measurements of typing fingers. **h** Device mounting positions on the typing hand. **i** IR images of the left-hand index finger under different typing postures. **j** Inconsistent temperature readings from IR camera during finger typing. **k** Temperature changes recorded by the device across various channels. **l**–**o** Real-time thermal measurements of climbers' fingers. **i** Device mounting positions on the climber's hands. The inset shows an IR image of the climber's hand. **m** Thermal mapping of normal and overused fingers on the right hand. **n** Comparison of Ch. 4 signal stability between conformal (yellow) and non-conformal (dangling, blue) electronics during climbing motion. **o** Monitoring temperature differences between the fingers of the left and right hands. Yellow and blue dashed lines indicate moments when the left or right hand is stretched, respectively. Most of the time, climbers are gripping handles. Source data are provided as a Source Data file.

We further evaluated the adaptability of the DAYS-fluid-transferred device on climbers' fingers (Fig. 4l–o). Climbing often subjects fingers to intense strain and overuse, leading to common injuries like inflammation of the tendons and ligaments[41]. Infrared (IR) imaging fails even during simple grasping because line-of-sight sensing is mechanically blocked (Fig. 4l inset). By contrast, devices transferred with DAYS fluid maintained distortion-free, intimate contact on the highly curved finger surface and provided stable real-time temperature readings with fluctuations below ±0.05 °C during both repeated grasp–release motions and actual climbing (Fig. 4m–o and Supplementary Fig. 15). The conformal sensor also detected localized temperature increases of 0.6–0.85 °C in overused fingers, demonstrating high sensitivity to inflammation-related thermal changes. Overuse was induced by repeatedly clenching the same hand for five minutes. In contrast, nonconformal sensors transferred using a high-viscosity fluid showed transient spikes, baseline drift, and failed to reliably distinguish the overused condition. Furthermore, Supplementary Fig. 16 presents continuous temperature monitoring on the bottom of the index finger, confirming stable signal output during long-term running conditions. These results confirm that DAYS fluid supports robust electronic transfer onto highly curved human surfaces, enabling reliable real-time physiological monitoring.

Beyond epidermal applications, we evaluated the versatility of the DAYS-fluid transfer process using ultrathin strain-sensor electronics (Supplementary Note 7 and Supplementary Fig. 17). On an extremely soft and fragile substrate such as lettuce leaf, DAYS-fluid enabled conformal transfer without tissue damage, preserving baseline resistance and yielding reliable strain-dependent signals (Supplementary Fig. h–j). In contrast, conventional transfer carriers, including elastomer-based (Supplementary Fig. c), water-based, and high-viscosity liquid methods (Supplementary Fig. d–g), failed to achieve strain-free integration, causing substrate folding, pattern distortion, or electrical instability. We further extended this demonstration to a hard, textured natural substrate (orange peel), where the strain sensor transferred by DAYS-fluid showed negligible resistance change and stable, reproducible responses under repeated bending, confirming distortion-free integration on rigid, uneven surfaces (Supplementary Fig. k–m).

## Discussion

In this study, we developed a deformation-decoupling, adhesion-switchable yield-stress fluid (DAYS-fluid) that enables reliable, conformal transfer of soft electronics onto complex, curved, and mechanically fragile biological surfaces. Composed of water and biocompatible fumed silica, DAYS-fluid exhibits stress-responsive rheology, transitioning reversibly from a solid-like to a fluidic state to facilitate distortion-free placement of delicate devices. By extensively tuning viscosity, we demonstrated that the fluid effectively decouples its deformation from the embedded electronics, minimizing strain during transfer to high-curvature geometries. We further showed that a water-triggered adhesion-switching mechanism significantly reduces interfacial adhesion, allowing clean and gentle detachment from diverse substrates. This dual functionality enables integration with previously inaccessible surfaces, including finger joints, mushroom caps, and even raw egg yolks. To validate practical utility, we achieved real-time temperature sensing on a human joint during dynamic motion. Taken together, this approach addresses longstanding challenges in the integration of electronics with soft, non-planar surfaces and opens new opportunities in wearable health monitoring, bioelectronics, and soft robotic systems.

## Methods

All human-subject procedures were conducted in accordance with the KAIST Institutional Review Board (IRB) and the Bioethics and Safety Act of the Republic of Korea. The study was classified and approved as an IRB-exempt, minimal-risk protocol.

One healthy adult male (34 years old) participated in the study. Sex was determined based on participant self-report. The participant provided written informed consent prior to participation and for publication of anonymized data and images. Participation was voluntary, and the participant could withdraw at any time without consequence. No financial compensation was provided.

In accordance with the SAGER guidelines, sex was documented in the study design. However, no sex- or gender-based comparative analysis was conducted because of the focus on device transfer mechanics rather than biological or physiological differences.

### Fabrication of ultrathin electrode arrays for conformal integration on curved substrates

Ultrathin electrode arrays were fabricated on a sacrificial planar substrate and subsequently transferred onto arbitrarily curved surfaces using a DAYS-fluid-mediated process. A thermally oxidized silicon wafer (SiO$_2$/Si) was employed as the base substrate. The wafer was first subjected to oxygen plasma treatment (150 W, 200 mtorr, 2 min) to enhance surface cleanliness and hydrophilicity, followed by sequential sonication in acetone, isopropanol, and deionized water for 5 min each. To minimize adhesion and facilitate subsequent release, the surface was functionalized with a monolayer of octadecyltrimethoxysilane (OTS; 3 mM in hexane) by spin-coating at 1500 rpm for 18 s and baking at 150 °C for 12 h.

A polyimide (PI) passivation layer (~3 μm) was formed by spin-coating a poly(pyromellitic dianhydride-co-4,4'-oxydianiline) amic acid precursor (Sigma-Aldrich) and curing at 250 °C for 1 h under ambient conditions. For electrical interconnects, a metal layer of either Cr/Au (10 nm/300 nm) or Cu (1 μm) was deposited via thermal evaporation or sputtering. Photolithography was used to define the overall electrode pattern. Subsequently, a second PI layer was spin-coated and cured at 150 °C for 12 h to encapsulate the metal traces.

Electrode patterns were further refined using a femtosecond laser micromachining system (EV Laser, High Precision FS LASER System). Unwanted laser-cut regions were mechanically removed using adhesive tape (Ease Release Tape, 3 M). The patterned film was then supported with a water-soluble tape (Wave Solder Tape 5414, 3 M) and transferred onto a PDMS-coated glass substrate. The PDMS layer was prepared by spin-casting Sylgard 184 (10:1 monomer-to-crosslinker ratio) and curing at 80 °C for 2 h.

To enable detachment and fluidic transfer, a sacrificial carrier layer of 10 wt% polymethyl methacrylate (PMMA; Mw ~39,500, Sigma-Aldrich) in chloroform was drop-cast onto the patterned electrode. Upon drying, the PMMA/electrode stack was delaminated and laminated onto the DAYS-fluid surface. The entire system was immersed in chloroform to selectively dissolve the PMMA layer, allowing the electrode array to settle and adhere onto the fluid interface after solvent evaporation in ambient air (~5 min).

The final transfer was accomplished by laminating the fluid-supported electrode array onto a curved target substrate. Conformal contact was ensured via mild lamination, and water was then applied to trigger adhesion switching at the fluid interface. This allowed for clean removal of the DAYS-fluid while leaving the electrode array intact and securely bonded to the curved substrate.

### Fabrication of thermal sensing wireless system

A flexible printed circuit board (fPCB) was developed to integrate an array of negative temperature coefficient (NTC) thermistors (4.7 kΩ, TDK Corporation). A rectangular region of the fPCB included open contact pads, allowing connection to a separate wireless-module fPCB via thin electrical wires. Each thermistor was mounted on serpentine-patterned sections of the same fPCB to preserve mechanical compliance and minimize constraints on natural body motion. A voltage-

divider circuit on the wireless module monitored the dynamic resistance of each thermistor, which was then digitized by a microcontroller unit (ISP 1807, Insight SIP) using its onboard analog-to-digital converter. The resulting 14-bit digital signals were subsequently transmitted via Bluetooth Low Energy (BLE) to an external device (for instance, an Apple iPad) for near-real-time visualization and data storage.

### Transfer and evaluation of conformal thermal sensing system on curved skin surfaces

Negative temperature coefficient (NTC) thermistors on the serpentine-patterned flexile PCB (f-PCB) were first encapsulated with a mechanically robust and thermally stable polyimide (PI) layer. A poly(-pyromellitic dianhydride-co-4,4'-oxydianiline) amic acid precursor solution (Sigma-Aldrich) was uniformly coated onto the NTCs and cured at 250 °C for 1 h under ambient conditions.

The encapsulated f-PCB was transferred onto a temporary polydimethylsiloxane (PDMS) substrate using a water-soluble adhesive tape (Wave Solder Tape 5414, 3 M), enabling ease of handling and precise alignment. To facilitate transfer onto the stress-responsive DAYS-fluid carrier, a sacrificial polymethyl methacrylate (PMMA) layer (10 wt% in chloroform) was applied by drop-casting and dried under ambient conditions. The PMMA-supported array was laminated onto the DAYS-fluid surface and subsequently printed onto a target curved substrate, specifically, the dorsal side of the finger joint, pre-coated with a thin layer of medical-grade adhesive (wig glue). Water was then introduced to trigger the adhesion-switching mechanism of the DAYS-fluid, allowing clean separation of the carrier fluid while leaving the thermistor-integrated device securely and conformally adhered to the skin.

To ensure reliable thermal sensing during motion, a compressible urethane foam layer (Ecoflex, Smooth-On) was applied over the thermistors to maintain intimate skin contact. Electrical wiring was completed by applying conductive silver paste (ELCOAT) to bond fine wires to the terminal pads of the f-PCB, followed by drying with a handheld hairdryer for 10 min. The exposed wiring was then encapsulated in soft siloxane rubber (Ecoflex, Smooth-On) to improve durability and user comfort.

For systematic performance validation, the sensing array was mounted on the left index finger joint of a healthy adult participant. Real-time resistance changes corresponding to skin temperature variations under different hand postures and distances were recorded using a Keithley 2635 A source measure unit and a Keithley 2450 source meter (Keysight Technologies). All experiments were conducted under ambient laboratory conditions.

For application-relevant scenarios such as typing and climbing, wireless data acquisition was enabled via a custom Bluetooth Low Energy (BLE) module integrated into the f-PCB, which was secured to the forearm using medical-grade adhesive tape. To emulate inflammation-like thermal changes, an overuse condition was induced by repeated clenching and unclenching of the fist for 5 minutes. Simultaneous thermal imaging with an infrared camera (FLIR T650SC, FLIR Systems, USA) provided a comparative benchmark to assess the sensing array's responsiveness and accuracy.

### Strain-sensor transfer and mechanical evaluation

Electrical connections between the strain sensor terminals and external wires were formed using conductive silver paste (ELCOAT). After application, the silver paste was dried using a handheld hairdryer for 10 min to ensure stable electrical contact. The exposed wiring and contact regions were subsequently encapsulated with a soft siloxane elastomer (Ecoflex, Smooth-On) to improve mechanical robustness, electrical insulation, and user comfort during deformation. Real-time

resistance signals of the strain sensors were recorded using a Keithley 2635 A source-measure unit and a Keithley 2450 source meter (Keysight Technologies). All electrical measurements were conducted under ambient laboratory conditions.

### Characterization of DAYS-Fluid for soft electronic transfer

To evaluate the rheological behavior of DAYS-fluid, shear-dependent and oscillatory measurements were performed using a rotational rheometer (Discovery HR-2, TA Instruments) equipped with a 20-mm parallel-plate geometry. A series of yield-stress fluids were prepared by dispersing various weight fractions of hydrophilic fumed silica into deionized water through high-speed mechanical stirring, followed by degassing to eliminate trapped air. The storage modulus (G'), loss modulus (G''), and shear viscosity were recorded as functions of shear stress and shear rate to determine the yield point and assess shear-thinning behavior, which is critical for conformal transfer printing.

The morphological features of the fumed silica and micro solid silica particles used in the fluid formulations were examined using transmission electron microscopy (TEM, JEM-2100F, JEOL), allowing precise analysis of particle size, shape, and network-forming potential within the fluid matrix.

Adhesion performance was quantitatively assessed through pull-off strength testing using a texture analyzer (Anton Paar). The samples were brought into conformal contact with PDMS or polyimide substrates under ambient conditions and subsequently detached to measure interfacial adhesion forces both before and after exposure to water, validating the efficacy of water-triggered adhesion switching.

Additionally, the optical clarity of the receiver substrates following fluid removal was examined using UV–Visible–NIR spectrophotometry (SolidSpec-3700, Shimadzu). Transmittance data across a broad wavelength range (200–2500 nm) were collected to evaluate residue-free detachment and surface cleanliness, further confirming the non-destructive nature of the transfer process.

### Reporting summary

Further information on research design is available in the Nature Portfolio Reporting Summary linked to this article.

## Data availability

All data supporting the results of this study are available within the paper and its Supplementary Information. Source data are provided with this paper.

## Code availability

No code was developed for this project.

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

## Acknowledgements

This work was supported by the National Research Foundation of Korea (NRF) funded by the Korean government (MSIT): RS-2024-00406240 and NRF-2021M3H4A1A02050234 (Y. S. Jung); 2022M3H4A1A01012816 and 2021R1A2C201071413 (J.-B. Yoon).

## Author contributions

K.S. and M.-K. Chung conceived the concept, designed the experiments, and performed the main fabrication and characterization. J.J. carried out femtosecond laser processing. J.P. provided application-specific measurement system. M.-U.K. assembled the flexible circuit system. J.-Y.Y. provided core methodologies for device-system integration. M.P. fabricated the serpentine device structures. G.L. designed and planned the electrode array experiments. J.H. operated water-soluble tape systems and other key experimental setups. J.-B.Y. and Y.S.J. supervised the overall project and provided critical guidance throughout the study. All authors discussed the results and approved the final manuscript.

## Competing interests

The authors declare no competing interests.
