## [Transparent Peer Review file · Nature Communications]

Deformation- and Damage-Free Transfer of Soft Electronics onto Highly Curved and Fragile Biological Surfaces

Corresponding Author: Professor Yeon Sik Jung

Version 0:

Reviewer comments:

Reviewer #1

(Remarks to the Author)

The manuscript introduces DAYS-fluid (Deformation-decoupling, Adhesion-switchable Yield-Stress fluid), a novel transfer medium for soft wearable and bioelectronics. By leveraging its unique stress-responsive rheological behavior, the authors achieve distortion-free placement of electronic devices onto complex, highly curved surfaces. Furthermore, DAYS-fluid exhibits water-triggered adhesion switching, enabling residue-free detachment from electronic devices. Compared to conventional transfer media or films, this system demonstrates superior control in both gripping and releasing devices while maintaining structural integrity. This is an exciting and impactful study. Therefore, I recommend the publication of this work in Nature Communications after addressing the following minor comments.

(1) Could the authors elaborate on the adhesion mechanism of DAYS-fluid with PDMS and PI? Since PDMS and PI possess very different surface energies and physicochemical properties, it would also be valuable to see adhesion tests with other soft substrates such as PU or SEBS.

(2) Water molecules appear to play a critical role in the adhesion-switching process. Does the intrinsic water content of DAYS-fluid also influence adhesion strength? It would be helpful to discuss this aspect, especially considering that DAYS-fluid may be hygroscopic. Before interfacial diffusion occurs, water molecules might preferentially diffuse into the bulk of the fluid, which could impact adhesion behavior.

(3) I am also curious whether this system is sensitive to external factors such as relative humidity and temperature. A discussion on these parameters would strengthen the manuscript.

Reviewer #2

(Remarks to the Author)

The manuscript introduces DAYS-fluid, a yield-stress fluid that combines deformation decoupling with water-triggered adhesion switching, aimed at transferring flexible electronics onto complex and fragile biological surfaces. The concept is novel and the demonstrations are interesting. However, there are several issues that limit the depth and clarity of the work, particularly in the framing of the introduction, the articulation of the main innovations, and the scope of the demonstrations. They should be addressed well before accepted. The detailed comments are below:

Major Comments

1. The introduction lacks a clear logical structure. Conformal contact is indeed a critical challenge, but the discussion is incomplete. The authors do not systematically outline the main routes to achieving conformal contact, nor do they explain whether transfer printing is the only viable approach. For transferring micro/nanostructures or electronic devices, what are the fundamental challenges? Without this broader framing, the unique contribution of DAYS-fluid is not sharply defined.
2. The paper positions thixotropic fluids as distinct, but in essence the mechanism is similar to sugar-transfer methods, which also rely on a solid-liquid transition. Sugar transfer (Science 378, 648–653 (2022) , <https://doi.org/10.1080/19475411.2020.1731622>) is already an established approach, but its key limitation lies in modulus mismatch, which compromises conformal integration with soft biological substrates. The introduction does not engage deeply with this prior work, making the innovation of DAYS-fluid less clearly articulated than it could be.
3. The adhesion switching mechanism relies on water-triggered decoupling, but this is not convincingly demonstrated in the applications. For epidermal electronics under long-term monitoring, sweat is an unavoidable factor that may influence

adhesion and performance, yet this scenario is not addressed or tested.

4. In the demonstration section, the comparison between conformal and non-conformal attachment is included, but the actual degree of performance difference is not fully explained.

5. Similarly, while the demonstrations show temperature sensing, the broader advantage of conformal adhesion is not only accuracy in thermal monitoring but also the reduction of motion artifacts. This potential is not convincingly demonstrated, though it could be one of the most impactful aspects of the technology.

6. Earlier results explore the limits of substrate hardness and softness for transfer, but the final demonstrations focus only on epidermal electronic devices. This does not fully highlight the advantages of the method.

7. Overall, the scientific contribution of the work is not in doubt, but the way it is written and presented does not do justice to its innovation and significance.

Minor Comments

1. On pages 4, the reported yield stress values are inconsistent, sometimes listed as 0.024 kPa and elsewhere as 0.025 kPa. It is also not clear whether the data are reproducible.

2. On page 11, the adhesion switching section still contains a “[REF]” placeholder.

3. In Fig. 1g, the geometric cases shown (①–⑥) are not clearly explained in the main text.

4. In Fig. 4e–f, the comparison between IR imaging and DAYS-fluid sensors lacks statistical support, such as sample size or error analysis.

Reviewer #3

(Remarks to the Author)

This manuscript presents a new method for transferring soft electronics onto complex surfaces by using the adhesion-switchable properties of a yield-stress fluid. The study is compelling and holds promise for printable ultrathin and on-skin electronics. However, although the authors claim superiority over conventional transfer-printing methods, the current data do not sufficiently substantiate this assertion. I recommend the following revisions to strengthen the work:

Please clarify the mechanism underlying the reversible phase transition. While the manuscript identifies the solid–liquid transition of the DAYS-fluid as critical, the current discussion and rheological data only demonstrate that a transition occurs, not why it occurs. Please provide a detailed mechanistic discussion.

Provide interfacial-level evidence of conformality and integrity. Fig. 1g shows optical images of transfers onto diverse substrates, but similar results can be achieved by conventional methods. To demonstrate clear advantages, include microscopic or cross-sectional imaging of the device–substrate interfaces on curved/undercut geometries, and quantify alignment error, voids, and delamination relative to benchmark carriers.

The demonstration focuses on thermistors. For modalities sensitive to interfacial impedance or strain (e.g., ECG/EMG, ECoG), does DAYS-fluid transfer preserve electrical performance and contact impedance versus standard methods? Benchmarking across modalities would broaden impact.

Include process videos. The water-assisted adhesion control and the transfer process are key innovations. Providing short videos of the transfer, adhesion switching, and residue-free removal would greatly improve clarity and reproducibility.

Correct minor formatting issues. For example, remove unwanted line breaks on pages 13 and 14.

Version 1:

Reviewer comments:

Reviewer #1

(Remarks to the Author)

The authors have addressed all of my comments, and I am satisfied. I recommend publication of this work in Nature Communications.

Reviewer #2

(Remarks to the Author)

I read through the point-by-point rebuttal and the revised manuscript carefully, and I appreciate the amount of work the authors put into addressing the comments. The revision clarifies the motivation and technical positioning much better, and the differences from existing transfer-printing approaches are now laid out more clearly, with additional experiments and analyses to back up the key claims. Overall, the main concerns I raised have been adequately resolved, and I do not have further questions.

I also went through the responses to the other two reviewers' comments, and I feel the authors handled those concerns well too, with thoughtful and convincing revisions supported by new data and clearer explanations. I'm satisfied with the current version and recommend acceptance.

List of responses and changes to the comments

Reviewer #1

General comments: General response

Comment 1: Response1, Response Fig. 1, Response References, Fig.3 c, Supplementary Note 4, Supplementary References

Comment 2: Response2, Response Fig. 2, Main text (page 12), Supplementary Note 6, Supplementary Fig. 8

Comment 3: Response 3, Response Fig. 3, Main text (page 12), Supplementary Note 7, Supplementary Fig. 9

Reviewer #2

General comments: General response

Major Comments

Comment 1: Response 1, Response Table 1 and 2, Response References, Main text (page 2), Supplementary Note 2, Supplementary Table 1 and 2, Supplementary References

Comment 2: Response 2, Response Table 3 and 4, Response References, Main text (page 2 and 4), Supplementary Table 3 and 4, Supplementary References

Comment 3: Response 3, Response Fig. 1-3, Main text (pages 11, 12 and 17), Fig. 3c, Supplementary Note 5, Supplementary Fig. 16, Supplementary References

Comment 4: Response 4, Response Fig. 4, Main text (page 16), Fig. 4d

Comment 5: Response 5, Response Fig. 5 and 6, Main text (page 16), Fig. 4n, Supplementary Fig. 14

Comment 6: Response 6, Response Fig. 6-8, Response References, Main text (page 6, 16, 18), Fig. 1f, Methods, References, Supplementary Note 8, Supplementary Fig. 17, Supplementary References

Comment 7: Response 7

Minor Comments

Comment 1: Response 1, Response Fig. 9, Main text (page 4), Supplementary Fig. 5

Comment 2: Response 2, Main text (page 13), References

Comment 3: Response 3, Main text (page 5), Fig. 1g

Comment 4: Response 4, Response Fig. 10, Main text (page 16), Fig. 4 e-g, Supplementary Fig. 12

Reviewer #3

General comments: General response

Comment 1: Response1, Response Fig. 1, Response Table 1, Reference References, Manuscript (page 3, 4, and 5), Supplementary Table 4, Supplementary Note 3, Supplementary Fig. 2, Supplementary references

Comment 2: Response2, Response Table 2, Response Fig. 2, Main text (page 6 and 9), Supplementary Note 4, Supplementary Table 5, Supplementary Fig. 7

Comment 3: Response 3, Response Fig. 3, Main text (page 17), Methods, Supplementary Fig. 17

Comment 4: Response 4, Supplementary Video 1, 2

Comment 5: Response 5

Reviewer #1

[General comments] *The manuscript introduces DAYS-fluid (Deformation-decoupling, Adhesion-switchable Yield-Stress fluid), a novel transfer medium for soft wearable and bioelectronics. By leveraging its unique stress-responsive rheological behavior, the authors achieve distortion-free placement of electronic devices onto complex, highly curved surfaces. Furthermore, DAYS-fluid exhibits water-triggered adhesion switching, enabling residue-free detachment from electronic devices. Compared to conventional transfer media or films, this system demonstrates superior control in both gripping and releasing devices while maintaining structural integrity. This is an exciting and impactful study. Therefore, I recommend the publication of this work in Nature Communications after addressing the following minor comments.*

[General response] We sincerely thank the reviewer for the positive evaluation of our manuscript and for recognizing the novelty and impact of our proposed DAYS-fluid system. We truly appreciate the reviewer's encouraging comments regarding the effectiveness and potential of this approach for distortion-free and residue-free transfer of soft electronics.

We have carefully addressed all of the reviewer's insightful suggestions, including additional discussion of the adhesion mechanism, expanded analysis of water-triggered switching, and inclusion of further data to improve clarity. We believe that these revisions have significantly strengthened the scientific depth and completeness of the manuscript.

[Comment 1] *Could the authors elaborate on the adhesion mechanism of DAYS-fluid with PDMS and PI? Since PDMS and PI possess very different surface energies and physicochemical properties, it would also be valuable to see adhesion tests with other soft substrates such as PU or SEBS.*

[Response 1] We thank the reviewer for this insightful comment. Although PDMS and PI have markedly different surface energies (typically $\sim 20 \text{ mJ m}^{-2}$ and $\sim 40\text{--}50 \text{ mJ m}^{-2}$, respectively), the adhesion of DAYS-fluid does not primarily arise from conventional surface-energy-driven wetting or chemical bonding mechanisms. Instead, it is governed by a **yield-stress-mediated mechanical interlocking process** that enables substrate-independent adhesion.

Upon contact, the viscoelastic DAYS-fluid deforms plastically to fill nanoscale asperities on the substrate surface, forming extensive physical interlocks that are dominated by normal stress and rheological flow, rather than surface tension. Because the yield stress ($\approx 0.025 \text{ kPa}$) is significantly lower than the modulus of typical substrates, the fluid can spontaneously conform to both rough and smooth surfaces regardless of their surface polarity, achieving consistent adhesion on substrates ¹.

To further validate this, we performed additional adhesion tests using polyurethane (PU, $35\text{--}48 \text{ mJ m}^{-2}$), styrene-ethylene-butylene-styrene (SEBS, $22\text{--}28 \text{ mJ m}^{-2}$), and glass substrates ($70\text{--}80 \text{ mJ cm}^{-2}$) (Response Fig. 1). All tests exhibited the same adhesion and detachment behaviors as observed with PDMS and PI: direct removal of the fluid led to cohesive tearing, while water-assisted detachment triggered a rapid adhesion switching behavior. These results confirm that the adhesion and release mechanism of DAYS-fluid is governed by its rheological characteristics, independent of the substrate type.

Response Fig. 1 | Substrate-independent adhesion and water-triggered detachment behavior of DAYS-fluid. Pull-off tests conducted on various substrates — PDMS (polydimethyl siloxane), PI (Polyimide, Kapton), PU (Polyurethane), SEBS (Styrene-Ethylene-Butylene-Styrene), and glass — demonstrate consistent adhesion and detachment characteristics regardless of surface chemistry or polarity. Under dry conditions, the DAYS-fluid exhibits cohesive tearing behavior due to strong mechanical interlocking with the substrate (“Tearing”). Upon water exposure, the adhesion rapidly decreases, allowing the fluid to delaminate cleanly from the surface (“Delaminating”).

Response References

1. Barral, Q., et al. Adhesion of yield stress fluids. *Soft Matter* **6**, 1343–1351 (2010).

[Manuscript]

A key advantage of this water-assisted mechanism is the ability to reduce the adhesion force to nearly zero, independent of substrate type. As shown in Fig. 3c, we measured the pull-off strength of the fluid on materials commonly used as adhesive and protective layers in soft electronic systems. In the absence of water, the fluid demonstrates significant pull-off strength, resisting detachment from the substrate. However, upon the introduction of water, the fluid detaches effortlessly, reducing pull-off strength to nearly zero. This substantial reduction in adhesion enables reliable and non-destructive transfer printing of electronics onto a variety of surfaces, greatly enhancing the practicality and efficiency of our system. – page 12, lines 15-8, 20-22

Importantly, as shown in Fig. 3c, this adhesion-switching behavior remains consistent across substrates with nearly threefold variation in surface energy³¹ including PDMS (~20–24 mJ m⁻²), PI (~40–47 mJ m⁻²), PU (~35–45 mJ m⁻²), SEBS (~22–25 mJ m⁻²), and glass (~65–75 mJ m⁻²). This substrate-independence arises because adhesion in DAYS-fluid is governed not by wetting or surface-energy-dependent interactions, but by yield-stress-mediated

mechanical gripping: the fluid plastically deforms into nanoscale asperities and forms mechanically interlocked contacts supported by its percolated silica network (Supplementary Note 5). – page 13, lines 28-32 and page 14, lines 1-2

Fig. 3 | Water-Assisted Adhesion Control for Clean Fluid Removal c, Variation in pull-off strength under dry and wet conditions across three different interfaces: PDMS -PI, fluid-PDMS, fluid-PI, fluid-PU, fluid-SEBS, and fluid glass.

Response References

31. Barral, Q., et al. Adhesion of yield stress fluids. *Soft Matter* **6**, 1343–1351 (2010).

Supplementary Note 5 | Substrate effect for Adhesion Mechanism of DAYS-fluid

DAYS-fluid adheres to substrates not through wetting or surface-energy-dependent interactions but through yield-stress-mediated mechanical gripping. Despite the $\sim 3\times$ variation in surface energy among PDMS ($\sim 20\text{--}24 \text{ mJ m}^{-2}$), SEBS ($\sim 22\text{--}25 \text{ mJ m}^{-2}$), PU ($\sim 35\text{--}45 \text{ mJ m}^{-2}$), PI ($\sim 40\text{--}47 \text{ mJ m}^{-2}$), and glass ($\sim 65\text{--}75 \text{ mJ m}^{-2}$), the measured pull-off strength remained nearly constant (Fig. 3c), indicating that adhesion does not correlate with substrate surface energy or wetting behavior.

Instead, adhesion arises from the percolated fumed-silica network, which plastically deforms under gentle pressure and interlocks with nanoscale surface asperities, generating mechanically supported contact points determined by the fluid's yield stress. This behavior aligns with the known adhesion characteristics of yield-stress fluids, where interparticle networks dominate interface mechanics rather than surface chemistry or wetting³⁰.

Upon water exposure, interfacial hydration locally disrupts hydrogen-bonded silica clusters, causing the mechanically supported interlocks to collapse and rapidly reducing adhesion to near zero. This switching process occurs regardless of substrate surface energy because adhesion is mechanically—not chemically—governed. These results confirm that DAYS-fluid operates through yield-stress-based gripping, enabling gentle, residue-free adhesion switching suitable for diverse biological and synthetic substrates.

Supplementary References

30. Barral, Q., et al. Adhesion of yield stress fluids. *Soft Matter* **6**, 1343–1351 (2010).

[Comment 2] *Water molecules appear to play a critical role in the adhesion-switching process. Does the intrinsic water content of DAYS-fluid also influence adhesion strength? It would be helpful to discuss this aspect, especially considering that DAYS-fluid may be hygroscopic. Before interfacial diffusion occurs, water molecules might preferentially diffuse into the bulk of the fluid, which could impact adhesion behavior.*

[Response 2] We thank the reviewer for this valuable comment. We fully agree that the intrinsic water content of DAYS-fluid plays a critical role in its adhesion-switching behavior. While the interfacial adhesion switching is primarily triggered by external water diffusion at the fluid–substrate interface, the internal hydration state of the fluid strongly influences how this interfacial process proceeds (Response Fig. 2).

When the water content was excessively high (Response Fig. 2a), the fluid exhibited a spreading morphology similar to that of pure water upon contact with the substrate. In this overhydrated state, water dominates interparticle interactions, weakening the fumed-silica network and leading to uncontrolled spreading across the surface. As a result, removal under dry conditions leaves behind residual traces of fluid due to wetting at the fluid–substrate interface, whereas removal under wet conditions causes the entire fluid to disperse into the surrounding water, leaving no recoverable film.

Conversely, when the water content was too low (Response Fig. 2c), the fluid adopted a dry, dough-like morphology. As previously discussed in Mechanistic basis of DAYS-fluid for adaptive contact behavior, underhydrated formulations crumble upon substrate contact, producing cracks and poor conformal wetting. In this regime, the fluid cannot establish sufficient interfacial contact or mechanical interlocking, resulting in weak adhesion and limited surface conformity. Even after exposure to water, these stiff, dough-like fluids maintain their shape and detach as a whole rather than undergoing smooth, selective interfacial separation.

These observations clarify that the efficiency and selectivity of the water-induced adhesion switching in DAYS-fluid depend not only on interfacial water diffusion but also on maintaining an appropriate internal hydration balance.

Response Fig. 2 | Effect of intrinsic water content on adhesion and detachment behavior of DAYS-fluid. Sequential schematic illustrations and optical images showing the adhesion and detachment behavior of the DAYS-fluid before and after water exposure: High water content (a), Optimal water content (b), Low water content (c).

[Manuscript]

Supporting this interpretation (Supplementary Note 6 and Supplementary Fig. 8), when the intrinsic water content of the fluid was intentionally increased, the overall hydrogen-bonding force between silica particles decreased, leading to weakened structural cohesion. Under these conditions, externally applied water readily diffused into the bulk rather than remaining interfacially confined, resulting in fluid dispersion instead of controlled interfacial detachment.

Conversely, under optimal hydration, interfacial diffusion was limited to the surface, enabling clean, residue-free adhesion switching. – page 13 lines 20-26

Supplementary Note 6 | Effect of Internal Water Content on the Adhesion-Switching Behavior of DAYS-fluid

To investigate the influence of intrinsic water content on the adhesion-switching performance of DAYS-fluid, a series of formulations with different internal hydration levels were systematically analyzed. Although the adhesion switching of DAYS-fluid is primarily triggered by external water diffusion at the fluid–substrate interface, the internal water content critically determines how this interfacial process progresses and whether the switching remains reversible and localized (Supplementary Fig. 10)

1. Overhydrated regime (particle content 0.08 wt.%)

When the water content of the fluid was excessively high, the DAYS-fluid exhibited a spreading morphology similar to that of pure water upon contact with the substrate (Fig. 1a, Supplementary Fig. 3b). In this overhydrated state, free water molecules dominate the interparticle interactions, weakening the hydrogen-bonded fumed silica network and reducing the yield stress. As a result, the fluid spreads uncontrollably across the surface and loses its cohesive integrity.

Under dry conditions, wetting at the fluid–substrate interface caused residual traces of fluid to remain on the surface. Under wet conditions, externally introduced water rapidly penetrated the bulk of the fluid, leading to complete dispersion and dissolution into the surrounding medium. This indicates that excessive internal water facilitates bulk disintegration rather than interfacial detachment, thereby undermining the controlled, reversible adhesion-switching process central to the DAYS-fluid mechanism.

2. Underhydrated regime (particle content 0.27wt.%)

When the internal water content was too low, the fluid exhibited a dry, dough-like morphology, as also shown in Fig. 1a and Supplementary Fig. 3b. Under these conditions, the interparticle hydrogen-bond network becomes overly rigid, preventing proper deformation and wetting. As previously discussed in *Mechanistic basis of DAYS-fluid for adaptive contact behavior*, such underhydrated formulations tend to crumble upon contact with the substrate, creating cracks and discontinuities at the interface. Consequently, adhesion remains weak and non-uniform, as the fluid cannot achieve sufficient conformal contact or mechanical interlocking. Even after exposure to water, these stiff, dough-like fluids maintain their structural rigidity.

3. Optimal hydration regime (particle content 0.11wt.%)

Between these two extremes lies an optimal hydration regime, in which the internal water content balances fluidity and structural cohesion. In this state, the hydrogen-bonded silica network remains continuous yet deformable, enabling controlled interfacial water diffusion and reversible adhesion switching. Water molecules introduced at the interface act as localized lubricants, promoting smooth, residue-free detachment without bulk deformation or dispersion.

Supplementary Fig. 8 | Effect of intrinsic water content on adhesion and detachment behavior of DAYS-fluid. Sequential schematic illustrations and optical images showing the adhesion and detachment behavior of the DAYS-fluid before and after water exposure: High water content (a), Optimal water content (b), Low water content (c).

[Comment 3] *I am also curious whether this system is sensitive to external factors such as relative humidity and temperature. A discussion on these parameters would strengthen the manuscript.*

[Response 3] We appreciate the reviewer's insightful question regarding the influence of environmental factors on the adhesion-switching behavior of DAYS-fluid. To address this, we systematically investigated the effects of relative humidity (RH) and temperature on the detachment dynamics under controlled conditions (Response Fig. 3).

The experiments were conducted by first placing DAYS-fluid droplets on slide glass substrates and exposing them to a constant environment inside a humidity and temperature chamber for 10 min. After this preconditioning, water was applied to the fluid surface, and the mobility of the droplets was evaluated by tilting the substrate to 60° and applying three mechanical taps, following the same protocol used in the main text (Fig. 3d–e). The moving head weight was recorded as an indicator of detachment progress.

(1) Effect of humidity: As shown in Response Fig. 3a, higher relative humidity accelerated the adhesion-switching process. Under humid conditions (RH > 70%), interfacial water molecules diffused more readily into the contact region, effectively lubricating the fluid–substrate interface and leading to faster detachment. Conversely, in dry environments (RH < 40%), limited ambient water slowed interfacial hydration, delaying adhesion switching.

(2) Effect of temperature: Temperature also played a critical role in determining detachment kinetics (Response Fig. 3b). When tested on an ice surface (3–5 °C), detachment occurred after approximately 3 min, reflecting the suppressed molecular mobility of water and reduced diffusion rate. In contrast, at 80 °C (hot plate condition), the fluid detached completely within 5 s, owing to enhanced interfacial diffusion and rapid disruption of the hydrogen-bond network.

These results confirm that DAYS-fluid maintains consistent adhesion-switching functionality across a wide range of environmental conditions, with detachment speed governed primarily by temperature- and humidity-dependent water mobility.

Response Fig. 3 | Effect of environmental humidity and temperature on the adhesion switching kinetics of DAYS-fluid. **a**, Detachment time of DAYS-fluid under different relative humidity (RH) conditions. Higher RH levels accelerate adhesion switching by facilitating interfacial hydration and promoting diffusion of water molecules across the contact interface. **b**, Temperature-dependent detachment behavior of DAYS-fluid. At elevated temperatures (80 °C), water diffusion and hydrogen-bond network disruption occur rapidly, resulting in complete detachment within 5 s. In contrast, at low temperatures (3–5 °C, on an ice surface), reduced molecular mobility of water delays detachment to approximately 3 min. Error bars indicate standard deviations from three independent measurements.

[Manuscript]

(3) With more than 20 seconds of exposure, the fluid underwent complete detachment, freely rolling over the surface (Fig. 3f). In addition, the adhesion-switching kinetics were also found to depend on environmental humidity and temperature, where enhanced interfacial water diffusion under humid or warm conditions accelerated detachment, whereas dry or low-temperature environments delayed it (Supplementary Note 7 and Supplementary Fig. 9). This signifies a near-total loss of interfacial adhesion, allowing the fluid to completely separate from the surface. – page 13, lines 1-235-37, lines 1-2

Supplementary Note 7 | Environmental Effects on Adhesion-Switching Behavior of DAYS-fluid

To examine the influence of environmental parameters on the adhesion-switching dynamics of DAYS-fluid, we performed a series of controlled tests varying both relative humidity (RH) and temperature conditions. The experiments were conducted using a temperature–humidity-controlled chamber.

DAYS-fluid droplets were first placed on a clean glass substrate and stabilized for 10 min under the target environmental condition (temperature: 3–80 °C; relative humidity: 30–80%). Following this preconditioning, a small volume of water (~100 μ L) was gently applied to the top of each droplet and then exposed to water to trigger adhesion switching. The mobility of the droplet was then evaluated by tilting the substrate to 60° and applying three mechanical taps, identical to the protocol described in the main text (Fig. 3d–e). The moving head weight (MHW), defined as the mass of the detached portion of the droplet, was measured over time to quantify the detachment progression. All tests were repeated three times for reproducibility.

Humidity dependence (Supplementary Fig. 3a): At high RH ($\geq 70\%$), water molecules readily diffuse into the interfacial region between the fluid and the substrate, rapidly disrupting the interfacial yield stress. This leads to fast adhesion loss and complete detachment within seconds after water exposure. Conversely, at low RH ($< 40\%$), insufficient environmental water vapor slows hydration at the interface, delaying adhesion switching. The critical role of ambient water availability suggests that DAYS-fluid exhibits a humidity-accelerated interfacial hydration process.

Temperature dependence (Supplementary Fig. 3b): Temperature modulates adhesion-switching kinetics by altering the diffusion rate and molecular mobility of interfacial water. Under low-temperature conditions (3–5 °C, on an ice surface), the hydrogen-bond network within the fluid remains stable, resulting in sluggish detachment (~3 min). At elevated temperatures (80 °C, hot plate), rapid water diffusion and hydrogen-bond rupture cause almost instantaneous detachment (~5 s).

Supplementary Fig. 9 | Effect of environmental humidity and temperature on the adhesion switching kinetics of DAYS-fluid. **a**, Detachment time of DAYS-fluid under different relative humidity (RH) levels. Higher RH accelerates adhesion switching by promoting interfacial hydration. **b**, Temperature-dependent detachment behavior. At 80 °C, rapid water diffusion and disruption of hydrogen-bond networks enable complete detachment within ~5 s, whereas at 3–5 °C reduced molecular mobility delays detachment to ~3 min. Error bars represent standard deviations from three independent measurements. Source data are provided as a Source Data file.

Reviewer #2

[General comments] *The manuscript introduces DAYS-fluid, a yield-stress fluid that combines deformation decoupling with water-triggered adhesion switching, aimed at transferring flexible electronics onto complex and fragile biological surfaces. The concept is novel and the demonstrations are interesting. However, there are several issues that limit the depth and clarity of the work, particularly in the framing of the introduction, the articulation of the main innovations, and the scope of the demonstrations. They should be addressed well before accepted. The detailed comments are below:*

[General response]

We deeply appreciate the reviewer's thorough and constructive feedback on our manuscript. We acknowledge that the reviewer raised important points regarding the logical structure of the introduction, the clarity of our innovation relative to prior work, and the scope of the demonstrations. In response, we have reorganized the Introduction to clarify the main challenges of achieving conformal contact, expanded comparisons with prior transfer techniques (including sugar-based methods), and added new experimental data and discussions on environmental effects, adhesion behavior, and performance differences. These revisions have improved the overall organization, clarity, and technical rigor of the manuscript.

Major Comments

[Comment 1] *The introduction lacks a clear logical structure. Conformal contact is indeed a critical challenge, but the discussion is incomplete. The authors do not systematically outline the main routes to achieving conformal contact, nor do they explain whether transfer printing is the only viable approach. For transferring micro/nanostructures or electronic devices, what are the fundamental challenges? Without this broader framing, the unique contribution of DAYS-fluid is not sharply defined.*

[Response 1] We sincerely thank the reviewer for this thoughtful and constructive comment. We agree that the logical framing of the *Introduction* can be improved to better define the unique contribution of our study.

Accordingly, we have reorganized the *Introduction* to clearly highlight that the novelty of our work lies in achieving conformal integration of semiconductor-grade microfabricated electronics onto highly curved and fragile biological surfaces. To clarify this contribution, our detailed response is structured into three parts: (1) routes to achieve conformal integration on curved biological surfaces, (2) advantages of transfer printing for semiconductor based high performance devices, and (3) fundamental challenges in transfer printing for high curvature biological surfaces. The corresponding revisions have been incorporated into both the main text and the Supplementary Information.

(1) Routes to achieve conformal integration on curved biological surfaces other than transfer printing

Five main strategies have been explored for achieving conformal integration of electronic devices on curved soft surfaces:

Holographic lithography: employs laser interference to directly define micro/nanostructures on 3D surfaces. It allows maskless, high-resolution patterning but is limited to photoresist-compatible materials and unsuitable for multilayer semiconductor devices.

Direct 3D printing (inkjet, aerosol jet, spray): enables additive patterning of conductive or functional inks directly on curved substrates but lacks the resolution, uniformity, and stability required for wafer-grade semiconductor devices.

Deformable substrates (stretchable and thermoforming): use elastomeric or thermoplastic substrates that can stretch or reshape to conform to curvature but experience strain-induced reliability issues and limited integration density.

Shape reconfiguration: relies on mechanical transformation (folding, buckling, or relaxation) of planar devices into 3D geometries, preserving wafer-level fidelity but restricted to pre-defined shapes and geometrically constrained curvatures.

Transfer printing: relies on the use of a temporary carrier that retrieves microfabricated devices from a donor substrate and deterministically places them onto a target surface through controlled adhesion modulation, enabling high fidelity integration of wafer grade electronics onto nonplanar geometries.

(2) **Advantages of transfer printing for semiconductor-based high-performance devices**

Microfabricated semiconductor devices offer substantial advantages when applied to bio-soft electronics, including exceptional electrical performance, long-term stability, and compatibility with multilayer, high-density architectures that are difficult to realize with printing-based or intrinsically stretchable materials. Their ultrathin geometries and mature fabrication processes enable precise signal acquisition and low-noise operation, which are essential for accurate monitoring of subtle physiological changes on soft and dynamically deforming tissues. When these devices can be conformally integrated onto biological surfaces without inducing mechanical strain or distortion, they provide a powerful platform for next-generation bioelectronic interfaces.

Transfer printing stands out as the only universal route for integrating wafer-grade, microfabricated semiconductor devices onto complex 3D or highly curved biological substrates. Unlike direct or deformable fabrication techniques, transfer printing decouples high-precision microfabrication from the mechanical placement process using a temporary carrier medium. This separation allows preservation of the original device integrity, electrical performance, and multilayer architecture fabricated on planar wafers, while achieving conformal contact with nonplanar, soft, or even biological targets. As summarized in Response Table 1, this approach uniquely combines high device fidelity, scalable integration, and biocompatibility—qualities not simultaneously achievable with any other method.

Response Table 1 | Comparison of representative strategies for conformal integration of electronic devices on curved biological surfaces. Five representative fabrication routes for integrating electronic devices onto curved biological surfaces are summarized. Each strategy is compared in terms of its working principle, compatibility with semiconductor devices.

Strategy	Schematics	Operating principle	Compatibility with semiconductor devices	Ref.
			 Accurate biological sensing and multifunctional integration.	
Holographic lithography		Uses laser interference patterns to define 3D or periodic microstructures on curved photoresist surfaces	Low Limited for multilayer semiconductor stacks or integrated circuits	1
Direct 3D printing (Inkjet, aerosol jet, spray)		Deposits conductive or functional inks directly onto curved substrates	Low Difficult to achieve multilayer or high-density integration	2-4
Deformable substrate (Stretchable, thermoforming)		Employs substrates that stretch or thermally deform to match surface curvature	Moderate Limited by mechanical strain and long-term reliability	5-8
Shape reconfiguration		Transforms planar devices into 3D geometries via mechanical relaxation, folding, or thermal shrinking	Moderate Geometry-dependent and less suitable for fragile biological surfaces	9-12
Transfer printing		Pre-fabricated devices are relocated via temporary carriers.	High Only universal route for integrating wafer-grade microdevices onto 3D curved biological substrates	13-20

(3) Fundamental challenges in transfer printing for high-curvature biological surfaces

Despite its promise, transfer printing has not yet been successfully demonstrated on highly curved biological surfaces. The main challenge lies in absence of the transfer carrier that satisfies the competing requirements of mechanical conformity and biological safety. At high curvature, large interfacial strain is easily imposed on the electronic device, risking fracture or delamination. Simultaneously, the underlying biological tissue can be thermally and mechanically damaged during either the conformal contact or detachment stages due to excessive stress or adhesion. Existing transfer media such as elastomeric stamps, adhesive films, and molten phase change carriers fail to achieve both the fidelity required for soft electronics and the safety required for biological surfaces.

This limitation motivated our development of DAYS-fluid, which uses stress-responsive phase transition, decouples mechanical deformation through an ultra-low yield stress, and achieves clean detachment through a water-triggered adhesion-switching mechanism, enabling non-destructive transfer of microfabricated semiconductor devices onto curved biological surfaces.

Response Table 2 | Comparative analysis of existing transfer printing carriers and the unmet need for an adaptive medium. This table summarizes representative transfer printing carriers, their contact-to-removal operating principles, and their applicability in terms of maintaining the fidelity of soft electronics on curved surfaces and ensuring the safety of fragile biological tissues.

Transfer printing carrier	Operating principle (Schematic: contact → removal)	Applicability		Ref.
		Fidelity of soft electronics on high curvature	Safety of fragile biological surface	
Elastomeric stamp (e.g., PDMS)	 Pick-up and release driven by pressure-induced kinetic adhesion and contact	Low Tensile strain causing misalignment of ultrathin devices.	Low Pressure stress inducing surface deformation	13-15
Adhesive-film	 Temporary adhesion switch using water-sensitive or UV-releasable films	Low Limited coverage of highly curved surfaces	High Gentle contact and clean removal	16
Thermal phase-change carrier (e.g., molten sugar)	 Solid-liquid transition triggered by heating and cooling cycles	High Conformal adaptation in molten state	Low Thermal damage	17-19
Hydrotransfer carrier	 Surface tension and capillary flow drive adhesion and release	Low Uncontrolled fluid motion causing pattern drift	High Gentle contact minimizing mechanical stress	20
DAYS-fluid (this work)	 Stress-responsive rheology with reversible solid-liquid transition and water-triggered adhesion switching	High Conformal, strain-free contact on complex curvatures	High Non-invasive transfer	

Response References

1. Purvis, A. *et al.* Photolithographic patterning of bihelical tracks onto conical substrates. *J. Micro/Nanolithogr. MEMS MOEMS* **6**, 043015 (2007).
2. Mohammed, M. G. & Kramer, R. All-printed flexible and stretchable electronics. *Adv. Mater.* **29**, 1604965 (2017).
3. Adams, J. J. *et al.* Conformal printing of electrically small antennas on three-dimensional surfaces. *Adv. Mater.* **23**, 1335–1340 (2011).
4. Carey, T., Jones, C., Le Moal, F., Deganello, D. & Torrisi, F. Spray-coating thin films on three-dimensional surfaces for a semitransparent capacitive-touch device. *ACS Appl. Mater. Interfaces* **10**, 19948–19956 (2018).
5. Ko, H. C. *et al.* A hemispherical electronic eye camera based on compressible silicon optoelectronics. *Nature* **454**, 748–753 (2008).
6. Plovie, B. *et al.* Arbitrarily shaped 2.5D circuits using stretchable interconnects embedded in thermoplastic polymers. *Adv. Eng. Mater.* **19**, 1–8 (2017).
7. Kim, R. H. *et al.* Waterproof AlInGaP optoelectronics on stretchable substrates with applications in biomedicine and robotics. *Nat. Mater.* **9**, 929–937 (2010).
8. Yang, Y. *et al.* 3D multifunctional composites based on large-area stretchable circuits with thermoforming technology. *Adv. Electron. Mater.* **4**, 1–10 (2018).
9. Rich, S. I., Lee, S., Fukuda, K. & Someya, T. Developing the nondevelopable: creating curved-surface electronics from nonstretchable devices. *Adv. Mater.* **34**, 2106683 (2022).
10. Chen, X. *et al.* Wrap-like transfer printing for three-dimensional curvy electronics. *Sci. Adv.* **9**, eadi0357 (2023).
11. Lin, C. *et al.* Highly deformable origami paper photodetector arrays. *ACS Nano* **11**, 10230–10235 (2017).
12. Cheng, Q. *et al.* Folding paper-based lithium-ion batteries for higher areal energy densities. *Nano Lett.* **13**, 4969–4974 (2013).

13. Meitl, M. A. *et al.* Transfer printing by kinetic control of adhesion to an elastomeric stamp. *Nat. Mater.* **5**, 33–38 (2006).
14. Sim, K. *et al.* Three-dimensional curvy electronics created using conformal additive stamp printing. *Nat. Electron.* **2**, 471–479 (2019).
15. Wang, Y. *et al.* Electrically compensated, tattoo-like electrodes for epidermal electrophysiology at scale. *Nat. Biomed. Eng.* **7**, 501–510 (2023).
16. Yan, Z. *et al.* Thermal release transfer printing for stretchable conformal bioelectronics. *Adv. Sci.* **4**, 1700251 (2017).
17. Le Borgne, B. *et al.* Conformal electronics wrapped around daily life objects using an original method: water transfer printing. *ACS Appl. Mater. Interfaces* **9**, 30345–30352 (2017).
18. Kim, D.-H. *et al.* Dissolvable films of silk fibroin for ultrathin conformal bio-integrated electronics. *Nat. Mater.* **9**, 511–517 (2010).
19. Giannakou, P., Tas, M. O., Le Borgne, B. & Shkunov, M. Water-transferred, inkjet-printed supercapacitors toward conformal and epidermal energy storage. *ACS Appl. Mater. Interfaces* **12**, 8456–8465 (2020).
20. Zabow, G. Reflow transfer for conformal three-dimensional microprinting. *Science* **378**, 648–653 (2022).

[Manuscript]

Recent progress in materials engineering and mechanical design has enabled semiconductor microfabricated bio-interfaced electronics to achieve stable signal acquisition and multimodal sensing, while maintaining intimate mechanical coupling with soft tissues. These advances now allow microfabrication-level precision to be applied directly on the skin for noninvasive, real-time physiological monitoring¹⁻³. Nevertheless, most device fabrication still depends on rigid, planar wafer substrates, making an additional integration step essential when interfacing with curved or mechanically fragile biological surfaces. Among existing integration strategies, transfer printing has emerged as the most versatile approach. It bridges

planar semiconductor microfabrication with three-dimensional biological targets by enabling precise and deterministic placement of ultrathin electronic structures onto nonplanar surfaces. Other routes, including in situ patterning, direct printing, and integration on thick stretchable substrates, have been explored, but they cannot directly accommodate microfabricated semiconductor devices, thereby limiting device complexity, resolution, and scalability (Supplementary Note 2, Supplementary Table 1)^{4–5}. – page 2, Lines 2-14

This need for an effective integration strategy becomes particularly pronounced when targeting convex, dynamically contoured biological surfaces where clinically relevant physiological signals are generated. Such geometries including finger and knee joints, retinal curvature, and various internal organs are mechanically active and highly three-dimensional, making them especially challenging for conformal electronic integration. Convex joints are especially prone to inflammation and structural degradation under repetitive mechanical loading, often resulting in localized changes in thermal and mechanical patterns that serve as early indicators of tissue damage⁶⁻⁷. – page 2, lines 17-21

Nevertheless, even state-of-the-art transfer printing techniques remain limited in their ability to reliably integrate electronics onto highly curved biological surfaces. A central challenge lies in the lack of suitable transfer carriers. During contact or release, existing media often impose mechanical or thermal stresses that can damage both the electronic devices and the underlying biological tissues (Supplementary Table 2). Traditional solid carriers, such as elastomers¹⁰⁻¹² and adhesive tape-based systems¹³, often deform electronic structures when applied to complex geometries, resulting in misalignment, poor adhesion, and compromised performance. – page. 2, Lines 31-34 and page 3, line 1

Supplementary Note 2 | Strategies for Conformal Integration on Curved Biological Surfaces

1 Existing approaches for integrating electronics onto curved biological surfaces (Supplementary Table 1)

A wide range of strategies has been developed to integrate flexible or stretchable electronic devices onto curved and dynamically contoured biological surfaces. Representative approaches include: (1) Halographic lithography, (2) Direct 3D printing (inkjet, aerosol jet, spray), (3) Deformable substrates (stretchable and thermoforming), (4) Shape reconfiguration. Although these strategies demonstrate varying degrees of conformal coverage, none can fully preserve the structural sophistication and alignment fidelity required for high performance electronic systems on challenging biological geometries.

2 Transfer printing as the only route compatible with semiconductor microfabrication (Supplementary Table 1)

In contrast to the above approaches, transfer printing provides the only broadly applicable route for integrating semiconductor microfabricated devices onto complex three dimensional biological surfaces. Its key advantage arises from decoupling high precision device fabrication from the integration process. Semiconductor devices can first be fabricated on planar wafers using established lithographic processes, ensuring exceptional electrical performance, submicron resolution, and multilayer architectural control. These devices are then detached from the wafer and deterministically transferred onto soft or curved biological targets using an intermediate carrier. Because transfer printing preserves the original microfabrication fidelity while allowing placement onto nonplanar surfaces, it achieves levels of complexity and performance that are unattainable with direct fabrication or printing based methods. As a result, transfer printing uniquely enables reliable integration of ultrathin semiconductor devices with strongly curved, fragile, or anatomically challenging biological surfaces.

Supplementary Table 1 | Comparison of representative strategies for conformal integration of electronic devices on curved biological surfaces. Five representative fabrication routes for integrating electronic devices onto curved biological surfaces are summarized. Each strategy is compared in terms of its working principle, compatibility with semiconductor devices.

Strategy	Schematics	Operating principle	Compatibility with semiconductor devices	Ref.
			 Accurate biological sensing and multifunctional integration.	
Holographic lithography		Uses laser interference patterns to define 3D or periodic microstructures on curved photoresist surfaces	Low Limited for multilayer semiconductor stacks or integrated circuits	1
Direct 3D printing (Inkjet, aerosol jet, spray)		Deposits conductive or functional inks directly onto curved substrates	Low Difficult to achieve multilayer or high-density integration	2-4
Deformable substrate (Stretchable, thermoforming)		Employs substrates that stretch or thermally deform to match surface curvature	Moderate Limited by mechanical strain and long-term reliability	5-8
Shape reconfiguration		Transforms planar devices into 3D geometries via mechanical relaxation, folding, or thermal shrinking	Moderate Geometry-dependent and less suitable for fragile biological surfaces	9-12
Transfer printing		Pre-fabricated devices are relocated via temporary carriers.	High Only universal route for integrating wafer-grade microdevices onto 3D curved biological substrates	13-20

Supplementary Table 2 | Comparative analysis of existing transfer printing carriers and the unmet need for an adaptive medium. This table summarizes representative transfer printing carriers, their contact-to-removal operating principles, and their applicability in terms of maintaining the fidelity of soft electronics on curved surfaces and ensuring the safety of fragile biological tissues.

Transfer printing carrier	Operating principle (Schematic: contact → removal)	Applicability		Ref.
		Fidelity of soft electronics on high curvature	Safety of fragile biological surface	
Elastomeric stamp (e.g., PDMS)	 Pick-up and release driven by pressure-induced kinetic adhesion and contact	Low Tensile strain causing misalignment of ultrathin devices.	Low Pressure stress inducing surface deformation	13-15
Adhesive-film	 Temporary adhesion switch using water-sensitive or UV-releasable films	Low Limited coverage of highly curved surfaces	High Gentle contact and clean removal	16
Thermal phase-change carrier (e.g., molten sugar)	 Solid-liquid transition triggered by heating and cooling cycles	High Conformal adaptation in molten state	Low Thermal damage	17-19
Hydrotransfer carrier	 Surface tension and capillary flow drive adhesion and release	Low Uncontrolled fluid motion causing pattern drift	High Gentle contact minimizing mechanical stress	20
DAYS-fluid (this work)	 Stress-responsive rheology with reversible solid-liquid transition and water-triggered adhesion switching	High Conformal, strain-free contact on complex curvatures	High Non-invasive transfer	

Supplementary References

1. Purvis, A. *et al.* Photolithographic patterning of bihelical tracks onto conical substrates. *J. Micro/Nanolithogr. MEMS MOEMS* **6**, 043015 (2007).
2. Mohammed, M. G. & Kramer, R. All-printed flexible and stretchable electronics. *Adv. Mater.* **29**, 1604965 (2017)
3. Adams, J. J. *et al.* Conformal printing of electrically small antennas on three-dimensional surfaces. *Adv. Mater.* **23**, 1335–1340 (2011).
4. Carey, T., Jones, C., Le Moal, F., Deganello, D. & Torrisi, F. Spray-coating thin films on three-dimensional surfaces for a semitransparent capacitive-touch device. *ACS Appl. Mater. Interfaces* **10**, 19948–19956 (2018).
5. Ko, H. C. *et al.* A hemispherical electronic eye camera based on compressible silicon optoelectronics. *Nature* **454**, 748–753 (2008).
6. Plovie, B. *et al.* Arbitrarily shaped 2.5D circuits using stretchable interconnects embedded in thermoplastic polymers. *Adv. Eng. Mater.* **19**, 1–8 (2017).
7. Kim, R. H. *et al.* Waterproof AlInGaP optoelectronics on stretchable substrates with applications in biomedicine and robotics. *Nat. Mater.* **9**, 929–937 (2010).
8. Yang, Y. *et al.* 3D multifunctional composites based on large-area stretchable circuits with thermoforming technology. *Adv. Electron. Mater.* **4**, 1–10 (2018).
9. Rich, S. I., Lee, S., Fukuda, K. & Someya, T. Developing the nondevelopable: creating curved-surface electronics from nonstretchable devices. *Adv. Mater.* **34**, 2106683 (2022).
10. Chen, X. *et al.* Wrap-like transfer printing for three-dimensional curvy electronics. *Sci. Adv.* **9**, eadi0357 (2023).
11. Lin, C. *et al.* Highly deformable origami paper photodetector arrays. *ACS Nano* **11**, 10230–10235 (2017).
12. Cheng, Q. *et al.* Folding paper-based lithium-ion batteries for higher areal energy densities. *Nano Lett.* **13**, 4969–4974 (2013).

13. Meitl, M. A. *et al.* Transfer printing by kinetic control of adhesion to an elastomeric stamp. *Nat. Mater.* **5**, 33–38 (2006).
14. Sim, K. *et al.* Three-dimensional curvy electronics created using conformal additive stamp printing. *Nat. Electron.* **2**, 471–479 (2019).
15. Wang, Y. *et al.* Electrically compensated, tattoo-like electrodes for epidermal electrophysiology at scale. *Nat. Biomed. Eng.* **7**, 501–510 (2023).
16. Yan, Z. *et al.* Thermal release transfer printing for stretchable conformal bioelectronics. *Adv. Sci.* **4**, 1700251 (2017).
17. Le Borgne, B. *et al.* Conformal electronics wrapped around daily life objects using an original method: water transfer printing. *ACS Appl. Mater. Interfaces* **9**, 30345–30352 (2017).
18. Kim, D.-H. *et al.* Dissolvable films of silk fibroin for ultrathin conformal bio-integrated electronics. *Nat. Mater.* **9**, 511–517 (2010).
19. Giannakou, P., Tas, M. O., Le Borgne, B. & Shkunov, M. Water-transferred, inkjet-printed supercapacitors toward conformal and epidermal energy storage. *ACS Appl. Mater. Interfaces* **12**, 8456–8465 (2020).
20. Zabow, G. Reflow transfer for conformal three-dimensional microprinting. *Science* **378**, 648–653 (2022).

[Comment 2] *The paper positions thixotropic fluids as distinct, but in essence the mechanism is similar to sugar-transfer methods, which also rely on a solid–liquid transition. Sugar transfer (Science 378, 648–653 (2022), <https://doi.org/10.1080/19475411.2020.1731622>) is already an established approach, but its key limitation lies in modulus mismatch, which compromises conformal integration with soft biological substrates. The introduction does not engage deeply with this prior work, making the innovation of DAYS-fluid less clearly articulated than it could be.*

[Response 2] We sincerely thank the reviewer for this thoughtful comment. We agree that the distinction between DAYS-fluid and the previously reported sugar-transfer technique was not clearly articulated in our original submission, which may have caused confusion for readers. To address this, we have prepared a detailed response focusing on (1) key differences relevant to applications on highly curved biosurfaces and (2) four fundamental technical distinctions between the two methods. These clarifications have been incorporated into both the Introduction and the Supplementary Note of the revised manuscript ¹⁻⁵.

(1) Key difference for applications on highly curved biological surfaces

- **Mechanical vs. thermal stimulus:** DAYS-fluid operates under mechanically applied, ultra-low stresses, whereas sugar-transfer requires thermal melting ($\geq \sim 60$ °C). The thermal route can induce thermal injury to biosurfaces, limiting applicability on fragile tissue.
- **Viscosity in operating state:** In the operating (liquid-like) state, DAYS-fluid exhibits low, stress-tunable effective viscosity, minimizing interfacial mechanical stress on microfabricated electronics. Molten sugar typically has higher, temperature-dependent viscosity, which can transmit deformation to devices and compromise alignment and integrity on tight curvatures.

Response Table 3 | Key mechanical implications of DAYS-fluid versus sugar-transfer for microdevice integration on highly curved biological surfaces. Comparison of DAYS-fluid and sugar-transfer in terms of their mechanical suitability for transferring microfabricated devices onto highly curved and fragile biosurfaces.

Category	Microdevices implications for highly curved bio-surfaces	
	DAYS-fluid (this work)	Sugar-transfer
Trigger stimulus	Mechanical (stress-responsive, ultra-low yield stress, ~tens of Pa)	Thermal (melting at ≥ 80 °C)
	Good Safe, non-invasive transfer on fragile biosurfaces (e.g., skin, egg yolk).	Bad Local heating or burns on soft tissues.
Viscosity In operating state	Low and tunable under applied stress	High and temperature-dependent
	Good Minimization of interfacial stress on microdevices	Bad Transmission of stress and distorts fragile electronics

(2) Four technical distinctions

(i) Trigger

- *DAYS-fluid*: **Stress-responsive**; activation by finely controlled, low mechanical stress (tens of Pa range), compatible with delicate biosurfaces (e.g., egg yolk, fingertip skin).
- *Sugar-transfer*: **Thermally responsive**; requires heating/cooling cycles for melting/recrystallization.

(ii) Phase-transition mechanism¹⁻⁴

- *DAYS-fluid*: **Yield-stress-mediated microstructural rearrangement** of a silica-water network (reversible break/reform of hydrogen-bonded clusters) \rightarrow solid-like \leftrightarrow liquid-like without molecular melting.
- *Sugar-transfer*: **Molecular phase change** (solid \leftrightarrow liquid) akin to ice-water transition.

(iii) Liquid-like / liquid-state behavior⁵

- *DAYS-fluid*: **Thixotropic, non-Newtonian**; viscosity decreases under shear and recovers at rest.
- *Molten sugar*: Predominantly **Newtonian**; viscosity governed by temperature, **insensitive to applied stress**, reducing fine control of interfacial mechanics.

(iv) Removal mechanism

- *DAYS-fluid*: **Surface-localized adhesion switching** triggered by interfacial water.
- *Sugar-transfer*: **Bulk dissolution** in water.

Response Table 4 | Four fundamental technical distinctions between DAYS-fluid and sugar-transfer. Schematic comparison of DAYS-fluid and sugar-transfer illustrating differences in activation stimulus, microstructural transition, rheological behavior, and detachment mechanism ¹⁻⁵.

Aspect	DAYS-fluid (this work)	Sugar-transfer
(i) Activation Trigger	 Stress-responsive Low mechanical stress	 Thermally responsive Heating and cooling cycles
(ii) Transition pathway	 Without melting Microstructure level Rearrangement within a silica-water network	 Melting Crystallization Molecular level Phase transition similar to ice-water transformation
(iii) Rheological behavior in the fluidic state	 Non-Newtonian Viscosity and yield behavior affected by applied stress	Newtonian Viscosity constant and independent of applied stress
(iv) Removal mechanism	 Surface-localized adhesion switching	 Bulk dissolution in water

Response References

1. Yuk, H. et al. 3D printing of conducting polymers. *Nat. Commun.* **11**, 1604 (2020).
2. Zhao, S. et al. Additive manufacturing of silica aerogels. *Nature* **584**, 387–392 (2020).
3. Sugino, Y. & Kawaguchi, M. Fumed and Precipitated Hydrophilic Silica Suspension Gels in Mineral Oil: Stability and Rheological Properties. *Gels* **3**, 32 (2017).
4. Kim, H. et al. Embedded Direct Ink Writing 3D Printing of UV Curable Resin/Sepiolite Composites with Nano Orientation. *ACS Omega* **8**, 23554–23565 (2023).
5. Arif, Z. N. et al. Designing and transforming yield-stress fluids. *Curr. Opin. Solid State Mater. Sci.* **23**, 100758 (2019).

[Manuscript]

Moreover, the mechanical stress required for conformal contact can destabilize both the electronics and the underlying biological tissues. Alternative carriers such as water bath–based hydro transfer and thermally activated materials including molten sugar¹⁴⁻¹⁷ offer improved conformability but involve complex handling procedures, limiting their practicality for bio-interfaced applications. can offer improved conformability. However, these approaches require complex handling procedures, which limit their practicality for bio-interfaced applications In particular, molten sugar requires thermal heating above 60 °C, and its intrinsically high liquid-state viscosity generates significant mechanical stress during transfer, which can damage fragile electronics and soft biological substrates. Such thermal cycling and the associated solid–liquid phase transitions render it unsuitable for soft or temperature-sensitive biological substrates as well as fragile electronic devices (Supplementary Table 3 and 4). These limitations underscore the need for a transfer medium capable of accommodating complex topographies while preserving both device integrity and biological compatibility—an essential step toward the seamless integration of electronics with fragile, anatomically challenging surfaces.– page 3, lines 5-14

This non-invasive capability is further demonstrated by its successful transfer onto egg yolks, which have an exceptionally low rupture pressure (0.2–0.5 kPa), far below the skin discomfort threshold (180-200 kPa) (Fig. 1f and Supplementary Fig. 6)²²⁻²⁷. These values illustrate that DAYS-fluid imposes negligible mechanical load on biological surfaces. In contrast, heat-based carriers such as molten sugar require melting and recrystallization at elevated temperatures (> 60 °C), which can generate thermal surface damage and high-viscosity–induced mechanical stress during operation. DAYS-fluid, by operating entirely under ambient conditions with an ultra-low yield stress, avoids these limitations and enables gentle, non-invasive transfer printing (Supplementary Tables 3 and 4).– page 5, lines 6-14

Supplementary Table 3 | Key mechanical implications of DAYS-fluid versus sugar-transfer for microdevice integration on highly curved biological surfaces¹⁻⁵. Comparison of DAYS-fluid and sugar-transfer in terms of their mechanical suitability for transferring microfabricated devices onto highly curved and fragile biosurfaces.

Category	Microdevices implications for highly curved bio-surfaces	
	DAYS-fluid (this work)	Sugar-transfer
Trigger stimulus	Mechanical (stress-responsive, ultra-low yield stress, ~tens of Pa)	Thermal (melting at ≥ 80 °C)
	Good Safe, non-invasive transfer on fragile biosurfaces (e.g., skin, egg yolk).	Bad Local heating or burns on soft tissues.
Viscosity In operating state	Low and tunable under applied stress	High and temperature-dependent
	Good Minimization of interfacial stress on microdevices	Bad Transmission of stress and distorts fragile electronics

Supplementary Table 4 | Four fundamental technical distinctions between DAYS-fluid and sugar-transfer. Schematic comparison of DAYS-fluid and sugar-transfer illustrating differences in activation stimulus, microstructural transition, rheological behavior, and detachment mechanism²¹⁻²⁵.

Aspect	DAYS-fluid (this work)	Sugar-transfer
(i) Activation Trigger	 Stress-responsive Low mechanical stress	 Thermally responsive Heating and cooling cycles
(ii) Transition pathway	 Without melting Microstructure level Rearrangement within a silica-water network	 Melting Crystallization Molecular level Phase transition similar to ice-water transformation
(iii) Rheological behavior in the fluidic state	 Non-Newtonian Viscosity and yield behavior affected by applied stress	Newtonian Viscosity constant and independent of applied stress
(iv) Removal mechanism	 Surface-localized adhesion switching	 Bulk dissolution in water

Supplementary Reference

21. Yuk, H. et al. 3D printing of conducting polymers. *Nat. Commun.* **11**, 1604 (2020).

22. Zhao, S. et al. Additive manufacturing of silica aerogels. *Nature* **584**, 387–392 (2020).
23. Sugino, Y. & Kawaguchi, M. Fumed and Precipitated Hydrophilic Silica Suspension Gels in Mineral Oil: Stability and Rheological Properties. *Gels* **3**, 32 (2017).
24. Kim, H. et al. Embedded Direct Ink Writing 3D Printing of UV Curable Resin/Sepiolite Composites with Nano Orientation. *ACS Omega* **8**, 23554–23565 (2023).
25. Arif, Z. N. et al. Designing and transforming yield-stress fluids. *Curr. Opin. Solid State Mater. Sci.* **23**, 100758 (2019).

[Comment 3] *The adhesion switching mechanism relies on water-triggered decoupling, but this is not convincingly demonstrated in the applications. For epidermal electronics under long-term monitoring, sweat is an unavoidable factor that may influence adhesion and performance, yet this scenario is not addressed or tested.*

[Response 3] We sincerely thank the reviewer for this thoughtful comment. This provides us with an opportunity to further clarify the role of the water-triggered adhesion-switching mechanism and its relevance to long-term epidermal monitoring.

The adhesion-switching behavior of the DAYS-fluid operates exclusively during the transfer stage, prior to physiological monitoring. After the electronic device is conformally transferred onto the biological surface, the fluid is completely removed, and the device adheres directly to the skin through its intrinsic adhesive or encapsulation layer. Therefore, perspiration generated during long-term monitoring does not influence the adhesion-switching mechanism itself (Response Fig. 1).

Response Fig. 1 | Schematic illustration of the water-triggered adhesion-switching mechanism of DAYS-fluid.

Instead, to demonstrate the universality of the adhesion behavior, we evaluated DAYS-fluid across substrate with widely varying surface energies. The fluid consistently exhibits switchable adhesion on substrates such as PDMS ($\sim 20\text{--}24 \text{ mJ m}^{-2}$), PI ($\sim 40\text{--}47 \text{ mJ m}^{-2}$), PU ($\sim 35\text{--}45 \text{ mJ m}^{-2}$), SEBS ($\sim 22\text{--}25 \text{ mJ m}^{-2}$), and glass ($\sim 65\text{--}75 \text{ mJ m}^{-2}$), demonstrating its versatility. As shown in Response Fig. 2, we measured the pull-off strength of the fluid under both dry and wet conditions across these substrates. In the dry state, measurable tearing strength was observed, whereas in the wet state, only the weight of the fluid was detected, and no measurable adhesion remained, confirming the effective water-triggered detachment (Response Fig. 2).

Response Fig. 2 | Substrate-independent adhesion and water-triggered detachment behavior of DAYS-fluid. Pull-off tests conducted on various substrates — PDMS (polydimethyl siloxane), PI (Polyimide, Kapton), PU (Polyurethane), SEBS (Styrene-Ethylene-Butylene-Styrene), and glass — demonstrate consistent adhesion and detachment characteristics regardless of surface chemistry or polarity. Under dry conditions, the DAYS-

fluid exhibits cohesive tearing behavior due to strong mechanical interlocking with the substrate (“Tearing”). Upon water exposure, the adhesion rapidly decreases, allowing the fluid to delaminating cleanly from the surface (“Delaminating”).

Additionally, to evaluate the potential influence of perspiration after device attachment, we performed real-time temperature monitoring using a thermal sensor array mounted on the underside of the index finger (Response Fig. 3). The participant’s finger was fully extended and secured to the treadmill handle to minimize environmental interference, and temperature was continuously recorded during sequential activity phases: a total of 30 minutes of running, with an additional 1 minute of measurement at the beginning and end of the session. A slight increase in skin temperature and visible perspiration were observed during the final running phase. The sensor maintained stable temperature readings throughout the test, confirming that DAYS-fluid–transferred electronics retain reliable performance even under mild perspiration conditions.

Response Fig. 3 | Real-time temperature monitoring using DAYS-fluid–transferred thermal sensors. (Left) Photograph of the real-time temperature sensing setup, where a thermal sensor array was conformally transferred to the underside of the index finger using DAYS-fluid. (Right) Real-time temperature monitoring recorded from channels (Ch. 1–Ch. 5) during sequential motion phases consisting of 1 min resting, 30 min running, and 1 min running. All channels maintain stable and continuous readings without baseline drift or signal spikes, confirming reliable operation even under mild perspiration conditions.

[Manuscript]

A key advantage of this water-assisted mechanism is the ability to reduce the adhesion force to nearly zero, independent of substrate type. As shown in Fig. 3c, we measured the pull-off strength of the fluid on materials commonly used as adhesive and protective layers in soft electronic systems. In the absence of water, the fluid demonstrates significant pull-off strength, resisting detachment from the substrate. However, upon the introduction of water, the fluid detaches effortlessly, reducing pull-off strength to nearly zero. This substantial reduction in adhesion enables reliable and non-destructive transfer printing of electronics onto a variety of surfaces, greatly enhancing the practicality and efficiency of our system. – page 12, lines 15-18, 20-22

Importantly, as shown in Fig. 3c, this adhesion-switching behavior remains consistent across substrates with nearly threefold variation in surface energy³¹ including PDMS (~20–24 mJ m⁻²), PI (~40–47 mJ m⁻²), PU (~35–45 mJ m⁻²), SEBS (~22–25 mJ m⁻²), and glass (~65–75 mJ m⁻²). This substrate-independence arises because adhesion in DAYS-fluid is governed not by wetting or surface-energy-dependent interactions, but by yield-stress-mediated mechanical gripping: the fluid plastically deforms into nanoscale asperities and forms mechanically interlocked contacts supported by its percolated silica network (Supplementary Note 5). – page 13, lines 28-32 and page 14, lines 1-2

Furthermore, Supplementary Fig. 16 presents continuous temperature monitoring on the bottom of the index finger, confirming stable signal output during long-term running conditions. These results confirm that DAYS fluid supports robust electronic transfer onto highly curved human surfaces, enabling reliable real-time physiological monitoring.– page 18, lines 5-9

Fig. 3 | Water-Assisted Adhesion Control for Clean Fluid Removal c, Variation in pull-off strength under dry and wet conditions across three different interfaces: PDMS -PI, fluid-PDMS, fluid-PI, fluid-PU, fluid-SEBS, and fluid glass.

References

31. Barral, Q., et al. Adhesion of yield stress fluids. *Soft Matter* **6**, 1343–1351 (2010).

Supplementary Note 5 | Adhesion Origin of DAYS-Fluid

DAYS-fluid adheres to substrates not through wetting or surface-energy-dependent interactions but through yield-stress-mediated mechanical gripping. Despite the $\sim 3\times$ variation in surface energy among PDMS ($\sim 20\text{--}24\text{ mJ m}^{-2}$), SEBS ($\sim 22\text{--}25\text{ mJ m}^{-2}$), PU ($\sim 35\text{--}45\text{ mJ m}^{-2}$), PI ($\sim 40\text{--}47\text{ mJ m}^{-2}$), and glass ($\sim 65\text{--}75\text{ mJ m}^{-2}$), the measured pull-off strength remained nearly constant (Fig. 3c), indicating that adhesion does not correlate with substrate surface energy or wetting behavior.

Instead, adhesion arises from the percolated fumed-silica network, which plastically deforms under gentle pressure and interlocks with nanoscale surface asperities, generating mechanically supported contact points determined by the fluid's yield stress. This behavior aligns with the known adhesion characteristics of yield-stress fluids, where interparticle networks dominate interface mechanics rather than surface chemistry or wetting³⁰.

Upon water exposure, interfacial hydration locally disrupts hydrogen-bonded silica clusters, causing the mechanically supported interlocks to collapse and rapidly reducing adhesion to near zero. This switching process occurs regardless of substrate surface energy because adhesion is mechanically—not chemically—governed. These results confirm that DAYS-fluid operates through yield-stress-based gripping, enabling gentle, residue-free adhesion switching suitable for diverse biological and synthetic substrates.

Supplementary Fig. 16 | Real-time temperature monitoring using DAYS-fluid-transferred thermal sensors. **a**, Schematic illustration of the water-triggered adhesion-switching process of DAYS-fluid. **b**, Photograph of the real-time temperature sensing setup, where a thermal sensor array was conformally transferred to the underside of the index finger using DAYS-fluid. **c**, Real-time temperature monitoring recorded from channels (Ch. 1–Ch. 5) during sequential motion phases consisting of 1 min resting, 30 min running, and 1 min running. All channels maintain stable and continuous readings without baseline drift or signal spikes, confirming reliable operation even under mild perspiration conditions. Source data are provided as a Source Data file.

Supplementary References

30. Barral, Q., et al. Adhesion of yield stress fluids. *Soft Matter* **6**, 1343–1351 (2010).

[Comment 4] *In the demonstration section, the comparison between conformal and non-conformal attachment is included, but the actual degree of performance difference is not fully explained.*

[Response 4] We thank the reviewer for this valuable comment. To clarify the quantitative difference between conformal and non-conformal attachment, we incorporated additional analysis and error-bar representations in the revised Fig. 4 (Response Fig. 4). In this experiment, participants performed repeated finger bending–extension motions at a fixed rate of one cycle every two seconds, and each measurement was conducted over three independent trials.

Specifically, when the DAYS-fluid–transferred device was conformally attached to the finger joint, the signal from channel 6 exhibited a very low standard deviation of 7.27 Ω , indicating stable, low-noise contact during motion. In contrast, the non-conformal (dangling) configuration showed a dramatically higher fluctuation with a standard deviation of 416.64 Ω , corresponding to a $> 50\times$ increase in resistance variance under identical conditions.

This substantial improvement highlights the critical role of conformal adhesion in maintaining consistent electrical coupling and minimizing motion-induced artifacts. The quantified comparison and updated error analysis have been added to the revised manuscript for clarity.

Response Fig. 4 | Real-Time Temperature Monitoring on Highly Curved Skin Surfaces. Real-time resistance changes of electronics with and without conformal transfer. Inset image shows the electronics without conformal transfer.

[Manuscript]

To validate the performance of the transferred device, we compared temperature sensing accuracy with and without conformal adhesion over three independent trials (Fig. 4d). The DAYS-fluid enabled seamless adhesion, resulting in accurate and consistent temperature readings during movement, with a low standard deviation of 7.27 Ω . In contrast, devices transferred using high-viscosity fluids exhibited poor adhesion, causing some channels to dangle and substantially degrading data reliability, as reflected by a much larger variation of 416.64 Ω . – page 17, lines 4-9

Fig. 4 | Real-Time Temperature Monitoring on Highly Curved Skin Surfaces. d, Real-time resistance changes of electronics with and without conformal transfer(n=3). Inset image shows the electronics without conformal transfer.

[Comment 5] *Similarly, while the demonstrations show temperature sensing, the broader advantage of conformal adhesion is not only accuracy in thermal monitoring but also the reduction of motion artifacts. This potential is not convincingly demonstrated, though it could be one of the most impactful aspects of the technology.*

[Response 5] We thank the reviewer for highlighting the importance of motion-induced artifacts in dynamic physiological monitoring. We agree that the ability to suppress motion artifacts represents a key advantage of conformal adhesion enabled by the DAYS-fluid.

In our demonstrations (Response Fig. 5-6), the DAYS-fluid–transferred thermal sensors maintained stable temperature readings during rapid and repeated movements such as typing and climbing. Even under continuous bending and stretching of finger joints, the recorded temperature traces showed no detectable spikes or baseline drift, indicating negligible motion artifacts. This result confirms that the strain-free conformal contact formed by the DAYS-fluid

provides robust signal stability during dynamic motions by preventing local delamination or strain mismatch between the device and the skin.

To further validate the impact of conformal adhesion on motion-artifact suppression, we performed additional control experiments using devices that lacked conformal contact. The dangling configuration was created using a high-viscosity fluid, which prevents strain-free lamination and leaves parts of the device suspended above the skin surface. Under this non-conformal condition, the temperature signals displayed significant fluctuations and transient spikes during both typing and climbing tasks. As shown in the typing control experiment (Response Fig. 5), the dangling device exhibited large oscillations of several degrees, whereas the conformally transferred device maintained a stable temperature trace with negligible variation. A similar trend was observed during climbing (Response Fig. 6), where repetitive gripping and release motions caused pronounced instability only in the dangling configuration. Moreover, the dangling configuration failed to resolve the subtle temperature increases associated with the overused finger, reflecting its limited contact fidelity and reduced sensing precision. These comparative results clearly demonstrate that the DAYS-fluid not only ensures high-fidelity temperature sensing but also provides reliable, motion artifact-free performance under dynamic body movement.

Response Fig. 5 | Comparison of temperature stability during typing motion under conformal and non-conformal attachment. Real-time temperature measurements recorded from channel 2 of the finger-mounted thermal sensor during repetitive typing motion. The

conformal condition (yellow), achieved using DAYS-fluid transfer, maintains a stable temperature profile with minimal fluctuation due to intimate contact with the skin. In contrast, the dangling condition (blue), produced by transfer with a high-viscosity fluid, exhibits pronounced fluctuations and transient deviations caused by partial detachment and motion-induced artifacts.

Response Fig. 6 | Real-time temperature monitoring on fingers of climber with conformal versus non-conformal (dangling) attachment. Real-time temperature profiles recorded from a DAYS-fluid-transferred thermal sensor (conformal, yellow) and a high-viscosity-transferred sensor (dangling, blue) during repeated gripping and release cycles. The conformal device maintains stable and high-fidelity temperature readings, clearly resolving localized temperature increases associated with overused fingers. In contrast, the dangling sensor exhibits large motion-induced fluctuations and baseline drift, and fails to distinguish the thermal signature of the overused finger due to intermittent contact and mechanical instability.

[Manuscript]

Our real-world tests further highlight the effectiveness of DAYS-fluid. In the first experiment, participant was equipped with a thermal sensing array mounted on the left index finger joint while performing a typing task (Fig. 4h-k). Repetitive tasks such as typing or writing can often lead to repetitive strain injury (RSI), causing inflammation of the ligaments and tendons⁴⁰. As shown in Fig. 4j, 4k and Supplementary Fig. 14, the conformally

transferred electronics provided continuous and low-artifact temperature signals on a moving finger, maintaining stable readings even during rapid finger motion. In contrast, sensor channels that failed to achieve intimate contact, as well as temperatures measured using IR imaging, exhibited continuous fluctuations and unstable behavior. – page 17, lines 22-26

Climbing often subjects fingers to intense strain and overuse, leading to common injuries like inflammation of the tendons and ligaments⁴¹. Infrared (IR) imaging fails even during simple grasping because line-of-sight sensing is mechanically blocked (Fig. 4l inset). By contrast, devices transferred with DAYS fluid maintained distortion-free, intimate contact on the highly curved finger surface and provided stable real-time temperature readings with fluctuations below ± 0.05 °C during both repeated grasp–release motions and actual climbing (Fig. 4m–o). The conformal sensor also detected localized temperature increases of 0.6–0.85 °C in overused fingers, demonstrating high sensitivity to inflammation-related thermal changes (Fig. 4m and n). Overuse was induced by repeatedly clenching the same hand for five minutes. In contrast, nonconformal sensors transferred using a high-viscosity fluid showed transient spikes, baseline drift, and failed to reliably distinguish the overused condition (Fig. 4n). These results confirm that DAYS fluid supports robust electronic transfer onto highly curved human surfaces, enabling reliable real-time physiological monitoring. – page 17, lines 30-34 and page 18, lines 1-9

Supplementary Fig. 14 | Comparison of temperature stability during typing motion under conformal and non-conformal attachment. Real-time temperature measurements recorded from channel 2 of the finger-mounted thermal sensor during repetitive typing motion. The conformal condition (yellow), achieved using DAYS-fluid transfer, maintains a stable temperature profile with minimal fluctuation due to intimate contact with the skin. In contrast, the dangling condition (blue), produced by transfer with a high-viscosity fluid, exhibits pronounced fluctuations and transient deviations caused by partial detachment and motion-induced artifacts.

Fig. 4 | Real-Time Temperature Monitoring on Highly Curved Skin Surfaces. n, Comparison of signal stability between conformal and non-conformal (dangling) electronics during climbing motion.

[Comment 6] *Earlier results explore the limits of substrate hardness and softness for transfer, but the final demonstrations focus only on epidermal electronic devices. This does not fully highlight the advantages of the method.*

[Response 6] We thank the reviewer for this insightful comment. We agree that demonstrating transfer robustness exclusively on epidermal devices may not fully convey the breadth of advantages offered by the DAYS-fluid carrier, especially regarding its ability to accommodate substrates with widely varying mechanical stiffness.

1. Rationale for focusing on epidermal devices

The primary objective of our main demonstration was to verify that DAYS fluid enables conformal and distortion free transfer of electronic devices onto highly curved biological surfaces. To represent this class of challenging geometries, we selected the finger joint as a model high curvature biological surface and employed a negative temperature coefficient (NTC) based thermal sensor array as the representative soft electronic platform.

The finger joint is a characteristic high curvature anatomical region with complex musculoskeletal structure, frequent mechanical loading, and highly thermos-mechanical sensitive skin (Response Fig. 6 and 7). Clinically, this site is particularly important. If semiconductor microfabricated electronics can be stably integrated onto the finger joint, they could enable three dimensional monitoring of inflammation, vascular alterations, and mechanical dysfunction. Despite this potential, most transfer printing demonstrations capable of handling microfabricated devices have focused on one dimensional strain sensors. The primary reason is that clinically relevant joint regions possess high curvature and soft, thermally sensitive skin, making it difficult for conventional transfer printing methods to safely and reliably place microfabricated devices. As shown in the high viscosity fluid demonstrations in Fig. 2c and Fig. 4d, elastomeric stamps, adhesive films, and stretchable substrates inevitably stretch or deform during application, imposing strain on the devices and resulting in signal acquisition. Hydrotransfer methods suffer from uncontrolled spreading at the water surface, whereas sugar based thermal carriers require elevated temperatures that pose risks of skin damage.

Among microfabricated electronic modalities, we selected an NTC based thermal sensor array as the representative device. This platform is fully fabricated using standard semiconductor microfabrication and incorporates multiple thermistors and interconnects distributed across a relatively broad area. Thermal sensors are highly sensitive to even slight detachment, making them a stringent benchmark for evaluating conformal integration quality. Moreover, because the finger joint exhibits skin folding and local overlap during extension, resistance temperature detectors (RTDs) tend to generate frequent artifacts, whereas a thermal sensor array composed of interconnects and localized temperature sensing nodes provides more robust operation with reduced susceptibility to mechanical noise.

Taken together, by achieving stable and artifact free temperature sensing across the full three dimensional contour of the finger joint, our demonstration highlights that DAYS fluid can integrate microfabricated soft electronics not only onto flat or mildly curved skin but also onto anatomically complex, highly curved, and clinically important biological surfaces.

Response Fig. 6 | Comparative plot summarizing the compressive pressures of yield stress of DAYS fluid, skin tactile sensitivity threshold, rupture pressure of egg yolk, skin pressure threshold for pressure injury (≥ 1 h exposure), skin pressure discomfort threshold, failure pressure of PDMS (Sylgard 184, 10:1), and perpendicular-to-grain failure pressure of oak hardwood.

Response Fig. 7 | Comparison of compressive pressure thresholds across biological and synthetic materials. Summary plot comparing representative compressive pressure levels for diverse biological and synthetic materials. Data points correspond to: the yield stress of DAYS-fluid, the skin tactile sensitivity threshold, the rupture pressure of egg yolk, the skin pressure threshold for pressure injury under ≥ 1 h exposure, the skin pressure discomfort threshold, the compressive failure pressure of PDMS (Sylgard 184, 10:1), and the perpendicular-to-grain compressive failure pressure of oak hardwood. These values illustrate the wide dynamic range of pressure tolerance across materials and highlight that the mechanical stress imparted by DAYS-fluid lies well below clinically relevant thresholds for sensitive biological tissues.¹⁻⁶

2. Additional validation on non-epidermal substrates of widely varying stiffness

As suggested by the reviewer, we performed additional experiments on two model substrates with extremely different mechanical stiffnesses to more clearly demonstrate the versatility and distinct advantages of the DAYS fluid (Response Fig. 8). For the soft substrate, we selected lettuce leaf, and for the hard substrate, we used orange peel. A strain sensor was chosen as the test device because it provides a stringent benchmark for evaluating the low-distortion transfer capability of the DAYS fluid.

The demonstration on the soft and fragile substrate, lettuce leaf, is presented in Response Fig. 8c-e. Similar to the results on orange peel, the DAYS fluid allowed gentle transfer onto the compliant lettuce surface without causing any physical damage, and the sensor resistance

showed negligible change before and after transfer (Response Fig. 8d). Because lettuce leaves lack structural support and easily deform under external forces, maintaining resistance integrity during transfer indicates that the yield-stress fluid exerts forces far below the threshold required to mechanically perturb or distort the leaf. Additionally, the transferred strain sensor produced stable strain signals even under small deformations (Response Fig. 8e), demonstrating reliable functionality on delicate biological tissues.

Response Fig. 8f-h illustrates the demonstration on the stiff natural substrate, orange peel. The sensor resistance remained unchanged before and after transfer (Response Fig. 8g), indicating that the DAYS fluid enabled conformal transfer onto the rigid, textured surface without cracking or delamination. Furthermore, during repeated bending of the orange peel, the transferred strain sensor exhibited stable and reproducible resistance modulation (Response Fig. 8h), confirming robust mechanical–electrical coupling even on rough, high-curvature surfaces.

Response Fig. 8 | Strain-sensor performance of DAYS-fluid-transferred devices on hard and soft biological substrates. **a**, Calibration curve of the ultrathin strain sensor, showing a linear relationship between bending-induced strain and relative resistance change ($\Delta R/R_0$). **b–d**, Strain-sensing behavior on a *hard* natural substrate (orange peel). Optical images of the sensor conformally laminated onto the orange peel (**b**). Comparison of sensor resistance before and after DAYS-fluid transfer, showing negligible change (**c**). Time-resolved resistance signals during repeated curvature changes of the orange peel, demonstrating stable and reproducible strain detection on a stiff, highly curved surface (**d**). **e–g**, Strain-sensing behavior on a *soft* biological substrate (lettuce leaf). Optical images of the sensor gently laminated onto the compliant lettuce surface (**e**). Resistance measured before and after transfer, confirming preserved device integrity (**f**). Dynamic resistance traces during low-amplitude deformation of the lettuce leaf, illustrating reliable sensing performance even on fragile, low-modulus substrates (**g**).

Response References

1. Johnson, K. O. & Phillips, J. R. *Tactile spatial resolution. I. Two-point discrimination, gap detection, grating resolution, and letter recognition.* **J. Neurophysiol.** **46**, 1177–1191 (1981).
2. Stadelman, W. J., Newkirk, D. & Newby, L. *Egg Science and Technology* (4th ed., Taylor & Francis CRC Press, 2017). ISBN 978-0-203-75887-8.
3. Reswick, J. B. & Rogers, J. E. Experience at Rancho Los Amigos Hospital with devices and techniques to prevent pressure sores. In: Kenedi, R. M. & Cowden, J. M. (eds) *Bed Sore Biomechanics* (Strathclyde Bioengineering Seminars, Palgrave, London, 1976).
4. Nasir, S. H., Troynikov, O., Wong Lit Wan, D. & Zheng, Z. Assessing the pressure and thermal discomfort thresholds for designing of therapeutic gloves: a pilot study. **OBM Integr. Complement. Med.** **4**, 3 (2019).
5. Hao, Y., Xie, J., Xu, B., Hu, B., Zheng, Y. & Shen, Y. Tunnel elasticity enhancement effect of 3D submicron ceramic (Al_2O_3 , TiO_2 , ZrO_2) fibers on polydimethylsiloxane (PDMS). **J. Adv. Ceram.** **10**, 502–508 (2021).
6. Carmona Uzcategui, M. G., Seale, R. D. & França, F. J. N. Physical and mechanical properties of clear wood from red oak and white oak. **BioResources** **15**, 4960–4971 (2020).

[Manuscript]

This non-invasive capability is further demonstrated by its successful transfer onto egg yolks, which have an exceptionally low rupture pressure (0.2–0.5 kPa), far below the skin discomfort threshold (180-200 kPa) (Fig. 1f and Supplementary Fig. 6)²²⁻²⁷. – page 5, lines 6-8

To validate the applicability of DAYS-fluid as a transfer medium for conformal integration of electronics onto convex, anatomically contoured biological surfaces, we implemented a thermal sensor array on the proximal interphalangeal joint of the index finger and conducted

real-time temperature monitoring during motion (Fig. 4a). The skin of the finger joint was chosen as a representative soft biological surface because it combines pronounced curvature with mechanical softness, making it one of the most challenging regions for conformal device integration. Indeed, conformal electronic mapping on this narrow anatomical site has rarely been demonstrated. – page 16, lines 5-9

Beyond epidermal applications, we evaluated the versatility of the DAYS-fluid transfer process using ultrathin strain-sensor electronics (Supplementary Note 8 and Supplementary Fig. 17). On an extremely soft and fragile substrate such as lettuce leaf, DAYS-fluid enabled conformal transfer without tissue damage, preserving baseline resistance and yielding reliable strain-dependent signals. In contrast, conventional transfer carriers, including elastomer-based, water-based, and high-viscosity liquid methods, failed to achieve strain-free integration, causing substrate folding, pattern distortion, or electrical instability (Supplementary Fig. 17c–g). We further extended this demonstration to a hard, textured natural substrate (orange peel), where the strain sensor transferred by DAYS-fluid showed negligible resistance change and stable, reproducible responses under repeated bending, confirming distortion-free integration on rigid, uneven surfaces (Supplementary Fig. 17k–m).– page 18, lines 11-18

Fig. 1 | Design features and operating principle of DAYS fluid. f, Comparative plot summarizing the compressive pressures of yield stress of DAYS fluid, skin tactile sensitivity threshold, rupture pressure of egg yolk, skin pressure threshold for pressure injury (≥ 1 h exposure), skin pressure discomfort threshold, failure pressure of PDMS (Sylgard 184, 10:1), and perpendicular-to-grain failure pressure of oak hardwood.

Methods

Strain sensor electrical connection and measurement

Electrical connections between the strain sensor terminals and external wires were formed using conductive silver paste (ELCOAT). After application, the silver paste was dried using a handheld hairdryer for 10 min to ensure stable electrical contact. The exposed wiring and contact regions were subsequently encapsulated with a soft siloxane elastomer (Ecoflex, Smooth-On) to improve mechanical robustness, electrical insulation, and user comfort during deformation. Real-time resistance signals of the strain sensors were recorded using a Keithley 2635A source-measure unit and a Keithley 2450 source meter (Keysight Technologies). All electrical measurements were conducted under ambient laboratory conditions.

References

22. Stadelman, W. J., Newkirk, D. & Newby, L. *Egg Science and Technology* (4th ed., Taylor & Francis CRC Press, 2017). ISBN 978-0-203-75887-8.
23. Reswick, J. B. & Rogers, J. E. Experience at Rancho Los Amigos Hospital with devices and techniques to prevent pressure sores. In: Kenedi, R. M. & Cowden, J. M. (eds) *Bed Sore Biomechanics* (Strathclyde Bioengineering Seminars, Palgrave, London, 1976).
24. Nasir, S. H., Troynikov, O., Wong Lit Wan, D. & Zheng, Z. Assessing the pressure and thermal discomfort thresholds for designing of therapeutic gloves: a pilot study. **OBM Integr. Complement. Med.** **4**, 3 (2019).
25. Hao, Y., Xie, J., Xu, B., Hu, B., Zheng, Y. & Shen, Y. Tunnel elasticity enhancement effect of 3D submicron ceramic (Al_2O_3 , TiO_2 , ZrO_2) fibers on polydimethylsiloxane (PDMS). **J. Adv. Ceram.** **10**, 502–508 (2021).
26. Carmona Uzcategui, M. G., Seale, R. D. & França, F. J. N. Physical and mechanical properties of clear wood from red oak and white oak. **BioResources** **15**, 4960–4971 (2020).

27. Catruño, C. et al. Psoriasis and Skin Pain: Instrumental and Biological Evaluations. *Acta Derm. Venereol.* **95**, 432–438 (2015).

Supplementary Fig. 6 | Comparison of compressive pressure thresholds across biological and synthetic materials. Summary plot comparing representative compressive pressure levels for diverse biological and synthetic materials. Data points correspond to: the yield stress of DAYS-fluid, the skin tactile sensitivity threshold, the rupture pressure of egg yolk, the skin pressure threshold for pressure injury under ≥ 1 h exposure, the skin pressure discomfort threshold, the compressive failure pressure of PDMS (Sylgard 184, 10:1), and the perpendicular-to-grain compressive failure pressure of oak hardwood. These values illustrate the wide dynamic range of pressure tolerance across materials and highlight that the mechanical stress imparted by DAYS-fluid lies well below clinically relevant thresholds for sensitive biological tissues.^{31–36} . Source data are provided as a Source Data file.

Supplementary Note 8. Strain-sensor-based evaluation of transfer-induced mechanical and electrical integrity

To evaluate whether the DAYS-fluid transfer process preserves electrical performance for sensing modalities sensitive to interfacial strain and mechanical instability, we performed comparative transfer experiments using ultrathin resistive strain sensors on substrates with markedly different mechanical properties (Supplementary Fig. 17). Strain sensors were

selected as a stringent benchmark because even minor transfer-induced deformation or interfacial instability immediately manifests as measurable electrical artifacts.

The strain sensor exhibits a linear and reproducible relationship between applied bending strain and relative resistance change ($\Delta R/R_0$), confirming its suitability for quantitative evaluation of transfer-induced mechanical effects (Supplementary Fig. 17a). We first examined transfer behavior on a highly compliant and fragile biological substrate (lettuce leaf), which represents a worst-case scenario for strain-sensitive electronics.

Consistent with the limitations of conventional transfer carriers discussed in Supplementary Fig. 17, existing transfer approaches induced significant mechanical and electrical degradation on the soft lettuce substrate. All conventional transfer methods were performed following the same procedures described in Supplementary Fig. xx to ensure a fair and consistent comparison. Elastomer-based carriers failed to achieve conformal contact because applied pressure compressed and folded the substrate, resulting in incomplete adhesion and transfer failure (Supplementary Fig. 17c). High-viscosity liquid-based carriers, employed as an analogue of molten-sugar transfer, transmitted shear and compressive stresses during contact, leading to visible stretching of the strain sensor and pronounced resistance changes after transfer (Supplementary Fig. 17d,e). Water-based hydrotransfer caused uncontrolled fluid flow during floating and substrate dipping, producing pattern distortion, positioning difficulty, and severe electrical instability (Supplementary Fig. 17f,g).

In contrast, DAYS-fluid enabled gentle, deformation-decoupled transfer on the same soft lettuce substrate. The strain sensor transferred using DAYS-fluid exhibited negligible resistance change before and after transfer and maintained stable, reproducible strain-resolved signals during low-amplitude deformation (Supplementary Fig. 17h–j). No parasitic strain, delamination, or baseline drift was observed, indicating robust interfacial coupling even on an extremely compliant biological surface.

To further confirm generality across mechanical regimes, we evaluated strain sensing on a hard, textured natural substrate (orange peel). Sensors transferred using DAYS-fluid showed unchanged resistance after transfer and stable, repeatable resistance modulation during repeated curvature changes (Supplementary Fig. 17k–m), demonstrating that the transfer process does not impose parasitic strain even on stiff, highly curved substrates.

Together, these results demonstrate that DAYS-fluid uniquely decouples transfer-induced mechanical stress from device integrity, preserving both mechanical and electrical performance across substrates spanning a wide stiffness range. This behavior directly supports the applicability of DAYS-fluid to strain- and impedance-sensitive bioelectronic modalities, beyond thermistors, where stable interfacial mechanics are essential.

Supplementary Fig. 17 | Comparative evaluation of conventional transfer-printing carriers and DAYS-fluid for strain-sensor integration on soft and hard substrates. a, Calibration curve of the ultrathin strain sensor. b, Photograph of a soft, fragile lettuce leaf substrate. c–g, Transfer onto lettuce leaf using conventional carriers: elastomer-based carriers cause substrate folding and transfer failure (c); high-viscosity liquid and water-based carriers induce sensor stretching or distortion with corresponding resistance instability (d–g). h–j, Strain-sensing behavior on a *soft* biological substrate (lettuce leaf). Optical images of the sensor onto the surface (h). Resistance measured before and after transfer (i). Dynamic resistance traces during low-amplitude deformation of the lettuce leaf. (j). k–m, Strain-sensing behavior on a *hard* natural substrate (orange peel). Optical images of the sensor onto the orange peel (k). Comparison of sensor resistance before and after DAYS-fluid transfer (l). Time-resolved

resistance signals during repeated curvature changes of the orange peel (m). Source data are provided as a Source Data file.

Supplementary References

31. Johnson, K. O. & Phillips, J. R. *Tactile spatial resolution. I. Two-point discrimination, gap detection, grating resolution, and letter recognition*. **J. Neurophysiol.** **46**, 1177–1191 (1981).
32. Stadelman, W. J., Newkirk, D. & Newby, L. *Egg Science and Technology* (4th ed., Taylor & Francis CRC Press, 2017). ISBN 978-0-203-75887-8.
33. Reswick, J. B. & Rogers, J. E. Experience at Rancho Los Amigos Hospital with devices and techniques to prevent pressure sores. In: Kenedi, R. M. & Cowden, J. M. (eds) *Bed Sore Biomechanics* (Strathclyde Bioengineering Seminars, Palgrave, London, 1976).
34. Nasir, S. H., Troynikov, O., Wong Lit Wan, D. & Zheng, Z. Assessing the pressure and thermal discomfort thresholds for designing of therapeutic gloves: a pilot study. **OBM Integr. Complement. Med.** **4**, 3 (2019).
35. Hao, Y., Xie, J., Xu, B., Hu, B., Zheng, Y. & Shen, Y. Tunnel elasticity enhancement effect of 3D submicron ceramic (Al_2O_3 , TiO_2 , ZrO_2) fibers on polydimethylsiloxane (PDMS). **J. Adv. Ceram.** **10**, 502–508 (2021).
36. Carmona Uzcategui, M. G., Seale, R. D. & França, F. J. N. Physical and mechanical properties of clear wood from red oak and white oak. **BioResources** **15**, 4960–4971 (2020).

[Comment 7] *Overall, the scientific contribution of the work is not in doubt, but the way it is written and presented does not do justice to its innovation and significance.*

[Response 7] We sincerely thank the reviewer for recognizing the scientific contribution of our work and for pointing out that the original presentation did not fully convey its innovation and significance. We have carefully revised the manuscript and Supplementary Information to strengthen the logical structure, improve clarity, and more explicitly highlight the conceptual advances of the DAYS-fluid platform. The main revisions are summarized below.

1. **Clearer conceptual framing in the Introduction (Major Response 1)** We revised the Introduction to more clearly position our work within existing conformal bio-integration strategies. The updated text summarizes major integration routes, clarifies why transfer printing is uniquely suited for wafer-fabricated devices on highly curved soft tissues, and identifies the central challenges of mechanical conformity and biosafety. These revisions appear in the main text and Supplementary Note 2.
2. **Mechanistic novelty of DAYS-fluid (Major Response 2)** We strengthened the mechanistic distinction between DAYS-fluid and sugar-based solid-liquid carriers by clarifying differences in activation trigger, viscosity, biosurface safety, and device strain. Four key distinctions—activation mechanism, phase-transition pathway, liquid-state rheology, and removal process—are now summarized in Supplementary Tables 3 and 4, emphasizing that DAYS-fluid is a fundamentally distinct transfer medium.
3. **Clarification of adhesion-switching mechanism (Major Response 3)** We clarified that water-triggered adhesion switching operates only during transfer and that the fluid is fully removed before physiological monitoring. Supporting schematics, pull-off tests, and new real-time temperature data collected during walking, resting, and running confirm the mechanism's physical basis and robustness.
4. **Strengthened demonstrations and relevance (Major Response 4-5)** We justified the finger joint as a stringent, clinically relevant testbed and showed that conformal adhesion significantly reduces motion artifacts. Dynamic tasks such as typing and climbing demonstrated that DAYS-fluid-transferred sensors provide stable signals, unlike non-conformal or IR-based measurements.
5. **Additional demonstrations on diverse substrates (Major Response 6)** To demonstrate generality beyond epidermal surfaces, we added experiments using ultrathin strain sensors on stiff (orange peel) and fragile (lettuce leaf) substrates. In both cases, resistance before

and after transfer remained unchanged and strain-resolved signals were stable, confirming distortion-free transfer across a wide stiffness spectrum.

6. **Improved quantitative rigor, reproducibility, and visualization (Minor Responses)**

We standardized yield-stress values, added sample sizes, mean \pm standard deviation, and error bars, and clarified the safety margin of DAYS-fluid using a comparative pressure plot.

Taken together, these revisions substantially strengthen the narrative, clarify the conceptual and mechanistic advances, and better connect the experimental demonstrations to the broader significance of DAYS-fluid for bio-integrated and soft electronics. We hope that the improved structure and presentation now do justice to the innovation and impact of the work, in line with the reviewer's concerns.

Minor Comments

[Comment 1] *On pages 4, the reported yield stress values are inconsistent, sometimes listed as 0.024 kPa and elsewhere as 0.025 kPa. It is also not clear whether the data are reproducible.*

[Response 1] We thank the reviewer for carefully pointing out this inconsistency. We have corrected the yield stress value throughout the manuscript for consistency and clarity. All references to the yield stress of DAYS-fluid have been unified to **0.0253 kPa**, which corresponds to the experimentally determined mean value from four independent measurements (mean \pm s.d. = 0.0253 ± 0.00135 kPa, $n = 4$). To further confirm reproducibility, we have included the corresponding data and statistical summary (Response Fig. 9) in the revised manuscript, showing excellent agreement among all samples with minimal variation.

Response Fig. 9 | Reproducibility of the yield stress measurements for DAYS-fluid. a. Storage modulus (G') as a function of applied shear stress for 0.11 wt.% DAYS-fluid, measured from four independent samples ($n = 4$). All samples exhibit consistent solid-like elasticity and a distinct yielding transition, where the storage modulus sharply decreases as the applied stress exceeds the yield point. **b,** Statistical summary of the yield stress values for the four samples, confirming high reproducibility across independent measurements (mean \pm s.d. = 0.0253 ± 0.00135 kPa, $n = 4$). These results demonstrate the consistency and stability of the rheological behavior of the 0.11 wt.% DAYS-fluid under ambient conditions.

[Manuscript]

Given the critical importance of viscosity control, further detailed in Fig. 2, a fluid formulation with a yield stress of $0.0253 (\pm 0.00135)$ kPa (Supplementary Fig. 5), achieved at a silica concentration of 0.11 wt.%, was selected for experimental validation. – page 5, lines 3-4

Supplementary Fig. 5 | Reproducibility of the yield stress measurements for DAYS-fluid.

a. Storage modulus (G') as a function of applied shear stress for 0.11 wt.% DAYS-fluid, measured from four independent samples ($n = 4$). All samples exhibit consistent solid-like elasticity and a distinct yielding transition, where the storage modulus sharply decreases as the applied stress exceeds the yield point. **b,** Statistical summary of the yield stress values for the four samples, confirming high reproducibility across independent measurements (mean \pm s.d. = 0.0253 ± 0.00135 kPa, $n = 4$). These results demonstrate the consistency and stability of the rheological behavior of the 0.11 wt.% DAYS-fluid under ambient conditions.

[Comment 2] *On page 11, the adhesion switching section still contains a “[REF]” placeholder.*

[Response 2] We appreciate the reviewer for catching this oversight. The “[REF]” placeholder in the adhesion-switching section has been replaced with the appropriate citation and supporting reference (page 11, lines 268–270).

[Manuscript]

This disparity likely arises from the lower water diffusion rate in polymer matrices and stronger adhesive interactions under wet conditions³²⁻³⁴. – page 14, line 14

References

32. Kim, G. Y. et al. Chiral 3D structures through multi-dimensional transfer printing of multilayer quantum dot patterns. *Nat. Commun.* **15**, 6996 (2024).

33. Nam, T. W. et al. Thermodynamic-driven polychromatic quantum dot patterning for light-emitting diodes beyond eye-limiting resolution. *Nat. Commun.* **11**, 3040 (2020).

34. Cho, S. H. et al. Selective, quantitative, and multiplexed surface-enhanced Raman spectroscopy using aptamer-functionalized monolithic plasmonic nanogrids derived from cross-point nano-welding. *Adv. Funct. Mater.* **30**, 2000612 (2020).

[Comment 3] *In Fig. 1g, the geometric cases shown (①–⑥) are not clearly explained in the main text.*

[Response 3] We thank the reviewer for pointing out the need for a clearer explanation of the geometric cases presented in Fig. 1g. In the revised manuscript, we have now explicitly described all six geometries corresponding to the figure—two finger joints with moderate convex curvature ($\tan \theta \approx 1$ to 3; Fig. 1g-①, ②), a sharp-edged rock with high convexity and discontinuous slopes ($\tan \theta \approx 5$ to 8; Fig. 1g-③), a dried fruit exhibiting irregular multi-curved wrinkles ($\tan \theta \approx 2$ to 6; Fig. 1g-④), a mushroom cap with negative Gaussian curvature and undercut geometry ($\tan \theta \approx -1$ to -6 ; Fig. 1g-⑤), and a quail egg yolk representing a soft, temperature-sensitive convex surface ($\tan \theta \approx 0.1$ to 0.3; Fig. 1g-⑥). This expanded description clarifies the distinct curvature profiles covered by each case and highlights how DAYS-fluid achieves conformal and damage-free transfer even on extreme or previously inaccessible geometries.

[Manuscript]

While previous studies primarily focused on low $\tan \theta$ (< 2) surfaces⁴⁻⁵, such as lens-shaped geometries, our findings demonstrate that DAYS-fluid extends beyond these conventional limits and successfully achieves conformal transfer even on surfaces exhibiting extreme curvatures and negative slopes—conditions previously unexplored in transfer printing. Specifically, DAYS-fluid demonstrates distortion-free transfer across a broad range of geometries with distinct curvature profiles: two finger joints with moderate convex curvature ($\tan \theta \approx 1$ to 3; Fig. 1g-①, ②), a sharp-edged rock exhibiting high convexity and

discontinuous slopes ($\tan \theta \approx 5$ to 8 ; Fig. 1g-③), a dried fruit featuring irregular multi-curved wrinkles ($\tan \theta \approx 2$ to 6 ; Fig. 1g-④), a mushroom cap with negative Gaussian curvature and undercut geometry ($\tan \theta \approx -1$ to -6 ; Fig. 1g-⑤), and a soft, hemispherical quail egg yolk with smooth curvature ($\tan \theta \approx 0.1$ to 0.3 Fig. 1g-⑥). These results highlight that DAYS-fluid enables conformal, damage-free device transfer on curvature regimes that were previously unattainable with conventional carriers, demonstrating its versatility across both rigid and deformable substrates. – page 6, lines 3-15

Fig. 1 | Design features and operating principle of DAYS fluid. g, Optical images of electronics transferred onto diverse substrates using DAYS-fluid: ① finger joint1, ② finger joint 2, ③ sharp-edged rock, ④ dried fruit, ⑤ mushroom cap, and ⑥ quail egg yolk. Source data are provided as a Source Data file.

[Comment 4] *In Fig. 4e–f, the comparison between IR imaging and DAYS-fluid sensors lacks statistical support, such as sample size or error analysis.*

[Response 4] We sincerely thank the reviewer for this valuable comment regarding the need for statistical analysis to support the comparison between IR imaging and DAYS-fluid-based sensors.

To address this point, we have added quantitative and statistical analyses to the revised manuscript, providing the sample size ($n = 3$), mean \pm standard deviation, and error bars for each condition corresponding to Fig. 4e–g. Specifically, three independent measurement cases were conducted under varying hand postures and camera distances, and the results were analyzed statistically. As shown in Response Fig.10, the DAYS-fluid–transferred sensors exhibit significantly lower standard deviations and smaller error ranges compared to IR imaging across all test conditions.

Response Fig. 10 | Statistical analysis corresponding to Fig. 4e–g: real-time thermal sensing under various hand postures and camera distances. a, c, e, Real-time temperature measurements obtained from three independent cases (Trial 2 and 3; $n = 3$) under varying hand postures (a, c) and different distances between the hand and IR camera (e). Each case represents an independent experimental condition showing consistent temperature profiles from DAYS-fluid–transferred sensors compared with IR imaging. **b, d, f,** Corresponding standard deviation plots of the temperature data (mean \pm s.d., $n = 3$), demonstrating significantly smaller fluctuations and improved stability in the DAYS-fluid sensors relative to IR imaging.

[Manuscript]

Additionally, these dangling devices often became entangled, contributing to user discomfort. Furthermore, as shown in Fig. 4e-g, and Supplementary Fig. 12, the thermal sensor array transferred by DAYS-fluid provided stable skin temperature readings, independent of posture, angle, or distance. In contrast, IR imaging showed errors exceeding 1.01 °C due to posture-dependent detection areas and failed to measure accurate skin temperature at greater distances where the readings increasingly reflected ambient temperature. – page 17, lines 9-15

Fig 4 | Real-Time Temperature Monitoring on Highly Curved Skin Surfaces. e-g, Real-time thermal measurements of the finger joint under varying hand postures (e and f) and distances from the camera (g). The upper left images display the experimental setup, while the upper right images show IR camera captures of the hand. The bottom graphs illustrate temperature changes recorded by the thermal sensor array and the IR camera.

Supplementary Fig. 12 | Statistical analysis corresponding to Fig. 4e–g: real-time thermal sensing under various hand postures and camera distances. a, c, e, Real-time temperature measurements obtained from three independent cases (Trial 2 and 3; $n = 3$) under varying hand postures (a, c) and different distances between the hand and IR camera (e). Each case represents an independent experimental condition showing consistent temperature profiles from DAYS-fluid-transferred sensors compared with IR imaging. **b, d, f,** Corresponding standard deviation plots of the temperature data (mean \pm s.d., $n = 3$), demonstrating significantly smaller fluctuations and improved stability in the DAYS-fluid sensors relative to IR imaging.

Reviewer #3

[General comments] *This manuscript presents a new method for transferring soft electronics onto complex surfaces by using the adhesion-switchable properties of a yield-stress fluid. The study is compelling and holds promise for printable ultrathin and on-skin electronics. However, although the authors claim superiority over conventional transfer-printing methods, the current data do not sufficiently substantiate this assertion. I recommend the following revisions to strengthen the work:*

[General response] We sincerely thank the reviewer for the thoughtful and constructive evaluation of our manuscript. We appreciate the reviewer's recognition that the proposed DAYS-fluid platform offers promising opportunities for the transfer of ultrathin and on-skin electronics. At the same time, we acknowledge the reviewer's request for stronger evidence to substantiate the claimed advantages. In response, we have substantially revised the manuscript and added new experiments, mechanistic analyses, and quantitative comparisons to address each of the reviewer's concerns.

[Comment 1] *Please clarify the mechanism underlying the reversible phase transition. While the manuscript identifies the solid–liquid transition of the DAYS-fluid as critical, the current discussion and rheological data only demonstrate that a transition occurs, not why it occurs. Please provide a detailed mechanistic discussion.*

[Response 1] We sincerely thank the reviewer for the insightful comment regarding the reversible phase-transition mechanism of DAYS-fluid. The solid–liquid transition observed in DAYS-fluid originates from the reversible formation and disruption of a percolated hydrogen-bond network between fumed silica nanoparticles and surrounding water molecules (Response Fig. 1)¹⁻⁶.

In the **solid-like state**, silanol (Si–OH) groups on the silica surface form dense, multivalent hydrogen-bond interactions with neighboring water molecules. These interactions generate a space-spanning, percolated microstructure that immobilizes the nanoparticles and endows the material with yield-stress–driven elasticity. In this state, the fluid behaves mechanically like a soft solid: it resists deformation under low stress, maintains its shape over time, and provides structural integrity capable of supporting lightweight electronic devices without flow or sagging. This behavior is analogous to jammed colloidal or gel-like materials in which particles are physically locked within a hydrogen-bonded network.

In the **liquid-like state**, when external shear or compressive stress exceeds the yield stress, the hydrogen-bond network is temporarily disrupted. The nanoparticles become unlocked from the jammed configuration and are free to reorganize along the direction of applied stress, enabling viscous flow. Under this condition, the material transitions into a deformable, shear-thinning fluid that can spread, adapt to underlying topographies, and conformally flow around curved or undercut geometries. Once the applied stress is removed, rapid re-association of water molecules with silanol groups restores the hydrogen-bond network, returning the system to its solid-like, load-bearing configuration without structural degradation^{1,2}.

This reversible stress-induced transition differs fundamentally from the conventional solid–liquid phase transition observed in materials such as water, where melting involves a thermodynamic change in molecular order. Instead, yield-stress fluids such as DAYS-fluid undergo a **mechanical** transition controlled by network rearrangement rather than thermal energy. This behavior is similar to widely known thixotropic systems, including toothpaste, paint, or cosmetic gels, which flow under shear but recover a solid-like structure at rest.

Response Fig. 1 | Mechanistic schematic of the phase transition in DAYS-fluid. The reversible solid–liquid transition of DAYS-fluid is governed by the dynamic hydrogen-bond network formed between fumed silica nanoparticles and water molecules. Resting state: In the absence of external stress, a percolated hydrogen-bonded network between silanol (Si–OH) groups and water molecules forms a jammed, solid-like microstructure exhibiting yield-stress behavior. Shear-induced flow state: When external stress exceeds the yield stress, the hydrogen-bond network is temporarily disrupted, allowing the nanoparticles to flow past one another and the fluid to behave in a viscous, deformable state.

To provide a more intuitive explanation, we additionally compared the solid–liquid transition characteristic of ordinary materials (e.g., ice or sugar melting) with the solid-like to liquid-like transition of yield-stress fluids, highlighting that the latter is governed by reversible microstructural reorganization rather than thermal phase change.

(i) Trigger: The activation mechanism of the two transfer media fundamentally differs. DAYS-fluid is stress-responsive and becomes activated under finely controlled, low mechanical stresses on the order of only a few tens of pascals, making it inherently compatible with delicate and mechanically fragile biosurfaces such as egg yolk or fingertip skin. In contrast, sugar-transfer methods are thermally responsive. They require heating and cooling cycles to induce melting and recrystallization, conditions that can expose biological tissues to elevated temperatures and introduce thermal stress during operation.

(ii) Phase-transition mechanism²⁻⁶: The phase-transition mechanism of DAYS-fluid is governed by yield-stress-mediated microstructural rearrangement within a silica-water network. Specifically, the reversible breaking and reforming of hydrogen-bonded clusters enable a transition between solid-like and liquid-like states without molecular melting. This stands in sharp contrast to sugar-transfer carriers, which rely on an actual molecular phase change between solid and liquid, analogous to the thermodynamic transition between ice and water.

(iii) Liquid-like / liquid-state behavior: In its liquid-like state, DAYS-fluid exhibits thixotropic, non-Newtonian behavior. Its viscosity decreases under applied shear stress and recovers when the stress is removed, enabling fine control of deformation and interfacial mechanics. Molten sugar, by comparison, behaves predominantly as a Newtonian fluid whose viscosity is governed by temperature rather than stress.

Response Table 1 | Four fundamental technical distinctions between DAYS-fluid and sugar-transfer. Schematic comparison of DAYS-fluid and sugar-transfer illustrating differences in activation stimulus, microstructural transition, rheological behavior, and detachment mechanism³⁻⁶.

Aspect	DAYS-fluid (this work)	Sugar-transfer
(i) Activation Trigger	 Stress-responsive Low mechanical stress	 Thermally responsive Heating and cooling cycles
(ii) Transition pathway	 Without melting Microstructure level Rearrangement within a silica-water network	 Melting Crystallization Molecular level Phase transition similar to ice-water transformation
(iii) Rheological behavior in the fluidic state	 Non-Newtonian Viscosity and yield behavior affected by applied stress	Newtonian Viscosity constant and independent of applied stress
(iv) Removal mechanism	 Surface-localized adhesion switching	 Bulk dissolution in water

Response reference

1. Teasell, R. W. et al. Cardiovascular consequences of loss of supraspinal control of the sympathetic nervous system after spinal cord injury. *Arch. Phys. Med. Rehabil.* **81**, 506–516 (2000).
2. Yuk, H. et al. 3D printing of conducting polymers. *Nat. Commun.* **11**, 1604 (2020).
3. Zhao, S. et al. Additive manufacturing of silica aerogels. *Nature* **584**, 387–392 (2020).
4. Sugino, Y. & Kawaguchi, M. Fumed and Precipitated Hydrophilic Silica Suspension Gels in Mineral Oil: Stability and Rheological Properties. *Gels* **3**, 32 (2017).
5. Kim, H. et al. Embedded Direct Ink Writing 3D Printing of UV Curable Resin/Sepiolite Composites with Nano Orientation. *ACS Omega* **8**, 23554–23565 (2023).
6. Arif, Z. N. et al. Designing and transforming yield-stress fluids. *Curr. Opin. Solid State Mater. Sci.* **23**, 100758 (2019).

[Manuscript]

Moreover, the mechanical stress required for conformal contact can destabilize both the electronics and the underlying biological tissues. Alternative carriers such as water bath-based hydro transfer and thermally activated materials including molten sugar¹⁴⁻¹⁷ offer improved conformability but involve complex handling procedures, limiting their practicality for bio-interfaced applications. can offer improved conformability. However, these approaches require complex handling procedures, which limit their practicality for bio-interfaced applications. In particular, molten sugar requires thermal heating above 60 °C, and its intrinsically high liquid-state viscosity generates significant mechanical stress during transfer, which can damage fragile electronics and soft biological substrates. Such thermal cycling and the associated solid–liquid phase transitions render it unsuitable for soft or temperature-sensitive biological substrates as well as fragile electronic devices (Supplementary Tables 3 and 4). These limitations underscore the need for a transfer medium capable of accommodating complex topographies while preserving both device integrity and biological compatibility—an essential

step toward the seamless integration of electronics with fragile, anatomically challenging surfaces.— page 3, lines 6-15

As illustrated in Fig. 1 b-d, in the absence of external stress, the fumed silica particles form a percolated network (Supplementary Note 3)¹⁴⁻¹⁷, imparting solid-like behavior, which securely immobilizes the electronics. In contrast to free-flowing liquids that generate uncontrolled currents misaligning delicate electronics⁷⁻⁸, this solid-like state facilitates precise placement of electronic devices onto the targeted skin location, before transfer. Upon application of stress, the silica particles align along the stress direction, inducing a reversible transition to a viscous liquid phase (Supplementary Note 3 and Supplementary Fig. 2)¹⁴⁻¹⁷. This stress-triggered fluidization allows seamless adaptation to a wide range of surface curvatures, making it particularly suitable for the integration of soft electronics with anatomically complex or highly contoured biological substrates. – page 4, lines 21 and 26

In contrast, heat-based carriers such as molten sugar require melting and recrystallization at elevated temperatures (> 60 °C), which can generate thermal surface damage and high-viscosity–induced mechanical stress during operation. DAYS-fluid, by operating entirely under ambient conditions with an ultra-low yield stress, avoids these limitations and enables gentle, non-invasive transfer printing (Supplementary Table 3 and 4). – page 5, lines 9-14

Supplementary Table 4 | Four fundamental technical distinctions between DAYS-fluid and sugar-transfer. Schematic comparison of DAYS-fluid and sugar-transfer illustrating differences in activation stimulus, microstructural transition, rheological behavior, and detachment mechanism²¹⁻²⁵.

Aspect	DAYS-fluid (this work)	Sugar-transfer
(i) Activation Trigger	 Stress-responsive Low mechanical stress	 Thermally responsive Heating and cooling cycles
(ii) Transition pathway	 Without melting Microstructure level Rearrangement within a silica–water network	 Melting Crystallization Molecular level Phase transition similar to ice · water transformation
(iii) Rheological behavior in the fluidic state	 Non-Newtonian Viscosity and yield behavior affected by applied stress	 Newtonian Viscosity constant and independent of applied stress
(iv) Removal mechanism	 Surface-localized adhesion switching	 Bulk dissolution in water

Supplementary Note 3 | Mechanism of rheological behavior in DAYS-fluid

The rheological behavior of the DAYS-fluid originates from the dynamic hydrogen-bond network between fumed silica nanoparticles and surrounding water molecules (Supplementary Fig. 2)²⁹⁻³¹.

In the **solid-like state**, silanol (Si–OH) groups on the silica surface form dense, multivalent hydrogen-bond interactions with neighboring water molecules. These interactions generate a

space-spanning, percolated microstructure that immobilizes the nanoparticles and endows the material with yield-stress–driven elasticity. In this state, the fluid behaves mechanically like a soft solid: it resists deformation under low stress, maintains its shape over time, and provides structural integrity capable of supporting lightweight electronic devices without flow or sagging. This behavior is analogous to jammed colloidal or gel-like materials in which particles are physically locked within a hydrogen-bonded network.

In the **liquid-like state**, when external shear or compressive stress exceeds the yield stress, the hydrogen-bond network is temporarily disrupted. The nanoparticles become unlocked from the jammed configuration and are free to reorganize along the direction of applied stress, enabling viscous flow. Under this condition, the material transitions into a deformable, shear-thinning fluid that can spread, adapt to underlying topographies, and conformally flow around curved or undercut geometries. Once the applied stress is removed, rapid re-association of water molecules with silanol groups restores the hydrogen-bond network, returning the system to its solid-like, load-bearing configuration without structural degradation.

This reversible stress-induced transition differs fundamentally from the conventional solid–liquid phase transition observed in materials such as water, where melting involves a thermodynamic change in molecular order. Instead, yield-stress fluids such as DAYS-fluid undergo a **mechanical** transition controlled by network rearrangement rather than thermal energy. This behavior is similar to widely known thixotropic systems, including toothpaste, paint, or cosmetic gels, which flow under shear but recover a solid-like structure at rest.

Supplementary Fig. 2 | Mechanistic schematic of the rheological behavior in DAYS-fluid.

The solid–liquid like transition of DAYS-fluid is governed by the dynamic hydrogen-bond network formed between fumed silica nanoparticles and water molecules. *Resting state*: In the absence of external stress, a percolated hydrogen-bonded network between silanol (Si–OH) groups and water molecules forms a jammed, solid-like microstructure exhibiting yield-stress behavior. *Shear-induced flow state*: When external stress exceeds the yield stress, the hydrogen-bond network is temporarily disrupted, allowing the nanoparticles to flow past one another and the fluid to behave in a viscous, deformable state.

Supplementary References

21. Yuk, H. et al. 3D printing of conducting polymers. *Nat. Commun.* **11**, 1604 (2020).
22. Zhao, S. et al. Additive manufacturing of silica aerogels. *Nature* **584**, 387–392 (2020).
23. Sugino, Y. & Kawaguchi, M. Fumed and Precipitated Hydrophilic Silica Suspension Gels in Mineral Oil: Stability and Rheological Properties. *Gels* **3**, 32 (2017).
24. Kim, H. et al. Embedded Direct Ink Writing 3D Printing of UV Curable Resin/Sepiolite Composites with Nano Orientation. *ACS Omega* **8**, 23554–23565 (2023).
25. Arif, Z. N. et al. Designing and transforming yield-stress fluids. *Curr. Opin. Solid State Mater. Sci.* **23**, 100758 (2019).
29. Yuk, H. et al. 3D printing of conducting polymers. *Nat. Commun.* **11**, 1604 (2020).

30. Barral, Q., et al. Adhesion of yield stress fluids. *Soft Matter* **6**, 1343–1351 (2010).

31. Johnson, K. O. & Phillips, J. R. *Tactile spatial resolution. I. Two-point discrimination, gap detection, grating resolution, and letter recognition. J. Neurophysiol.* **46**, 1177–1191 (1981).

[Comment 2] *Provide interfacial-level evidence of conformality and integrity. Fig. 1g shows optical images of transfers onto diverse substrates, but similar results can be achieved by conventional methods. To demonstrate clear advantages, include microscopic or cross-sectional imaging of the device–substrate interfaces on curved/undercut geometries, and quantify alignment error, voids, and delamination relative to benchmark carriers.*

[Response 2] We thank the reviewer for this insightful comment highlighting the need for interfacial-level evidence and quantitative benchmarking against conventional transfer-printing carriers. To directly address this point, we performed a systematic comparative study on a highly compliant and undercut substrate—the mushroom cap shown in Fig. 1g-⑤—which represents one of the most challenging geometries for conformal electronic integration (Response Table 2).

We compared elastomer-based carriers, water-based carrier (hydrotransfer), high-viscosity liquid carriers (molten-sugar analogue), and the proposed DAYS-fluid under identical transfer conditions. To move beyond surface-level optical inspection, we additionally acquired magnified confocal microscopy images to directly visualize the device–substrate interface, enabling assessment of local detachment and deformation on curved regions.

To quantitatively evaluate interfacial integrity, we introduced two complementary interfacial error metrics: detached area (%) and geometric strain (%). Detached area was defined as the fraction of the total device area that failed to adhere to the mushroom substrate. Geometric

strain was quantified by analyzing deformation of serpentine interconnects. For each curved segment, magnified optical microscopy images were compared with the original device design, and the apparent arc radius and angular distortion were extracted to estimate the local strain and calculate an average value. The degree of distortion (strain) was evaluated based on established correlations between serpentine geometry deformation and applied strain reported in prior literature¹.

Below, we describe the characteristic failure modes observed for each representative carrier and relate these observations to their underlying mechanical and fluidic transfer mechanisms.

(1) Elastomer-based carriers (stamps, pads, stretchable substrates)

Elastomer-based transfer carriers require full-area, uniform contact between the device and the target substrate to achieve stable adhesion during transfer. However, for highly compliant and undercut substrates such as mushroom tissue, the applied pressure from the elastomer predominantly compresses the substrate itself rather than promoting conformal contact with the device. As observed in the optical images, this substrate compression prevents uniform device–substrate contact, leading to partial adhesion or complete transfer failure. To visually assess the contact interface, an ultrathin PDMS film was used in the experiments, which clearly revealed the lack of full-area contact. When higher pressure was applied to improve adhesion, mechanical damage and fracture of the mushroom substrate were consistently observed.

(2) Water-based carriers (hydrotransfer printing)

In conventional hydro-dipping approaches, pattern spreading is suppressed by supporting the electronics on a carrier film. However, this configuration is incompatible with highly curved substrates, which require substrate-free electronics to achieve conformal wrapping. In the absence of a carrier film, lateral water flow induces uncontrolled pattern spreading. For curved substrates in particular, although the initial contact point may appear stable, continued insertion of the

substrate generates fluid motion along the curvature. This flow results in additional strain, misalignment, and loss of pattern fidelity, as observed in the optical images.

(3) High-viscosity liquid-based carriers (molten-sugar analogue)

High-viscosity liquid-based carriers, such as molten sugar, rely on bringing the electronic device into contact with the substrate through a viscous liquid phase. Because direct implementation of molten sugar transfer is experimentally challenging, we employed a high-viscosity fluid composed of silica particles as a practical analogue. As shown in the images, the electronics undergo pronounced stretching when transferred onto the mushroom substrate. This behavior originates from the deformation of the viscous fluid during contact, which transmits shear and compressive stresses to the electronic device, as described in Fig. 2.

(4) DAYS fluid (This work)

In contrast to the conventional carriers described above, DAYS-fluid enables conformal, strain-free transfer on highly compliant and undercut substrates such as mushroom tissue. Owing to its viscosity-controlled, stress-responsive flow behavior, DAYS-fluid adapts locally to complex surface topographies without compressing the substrate or transmitting shear stress to the device.

Response Table 2 | Comparative feasibility and interfacial integrity of transfer-printing carriers on highly compliant, undercut substrates. Photographic and confocal microscopy–based evaluation of representative transfer-printing carriers, including elastomer-based carriers (stamps, pads, stretchable substrates), water-based hydrotransfer, high-viscosity liquid carriers (molten-sugar analogue), and the DAYS–fluid developed in this work.

		Elastomer (ex. Stamp, Pad, Stretchable substrate)	Water	High viscosity liquid (ex molten sugar)	This work
Optical image							Not Feasible	Not feasible	Not feasible	Feasible
Confocal Microscopy																	Not Feasible	Not feasible	Not feasible	Feasible
Mechanism		Require full-area contact for adhesion Substrate compression causes non-uniform contact or damage	Uncontrolled fluid flow during floating or dipping	Viscous flow transmits shear and compressive stress	Detailed rheology control enables conformal contact
Quantified interfacial error	Detached area (%)	-	4.75%	8.14%	0%
	Geometric Strain (%)	-	2.08%	5.27%	0%

Response Fig. 2 | Quantification of geometric strain as an interfacial error metric. Confocal microscopy (OM) images showing representative electrode patterns transferred using a high-viscosity fluid (left) and DAYS-fluid (right). Geometric strain was quantified by comparing the local curvature and arc geometry of the transferred serpentine traces with the original device design. For each curved segment, the apparent arc angle and radius were extracted from OM images and referenced to the undeformed geometry, enabling estimation of transfer-induced geometric strain (yellow guides).

References

1. Y. Sun and W. G. Chong, *Mater. Horiz.*, **10**, 2373-2397 (2023)

[Manuscript]

To further substantiate the interfacial conformality of DAYS-fluid, we compared multiple transfer media on a mushroom substrate with negative Gaussian curvature and undercut geometry. Magnified optical and confocal images (Fig. 1g-⑤, Supplementary Note 4, Supplementary Table 5, and Supplementary Fig. 7) reveal that conventional carriers cause interfacial distortion, folding, or partial delamination, leading to non-uniform contact and strain accumulation. In contrast, DAYS-fluid achieves uniform, strain-free conformal contact across the entire interface, including undercut regions. – page 6 lines 22-28

By precisely minimizing the viscosity of DAYS-fluid, we achieved effective decoupling of carrier deformation from the electronics, thereby mitigating mechanical constraints typically encountered during the transfer process (Fig. 2a-c, Supplementary Note 4, Supplementary Table 5, and Supplementary Fi. 7). – page 9, line 4-5

Supplementary Note 4 | Interfacial-level comparison of transfer-printing carriers on highly compliant and undercut substrates

To provide interfacial-level evidence of conformality and integrity, we performed a systematic comparative study using a highly compliant and undercut biological substrate: the mushroom cap shown in Fig. 1g-⑤. This geometry represents one of the most challenging cases for conformal electronic integration due to its low stiffness, negative curvature, and undercut morphology. Elastomer-based carriers, water-based hydrotransfer, high-viscosity liquid carriers (molten-sugar analogue), and the proposed DAYS-fluid were compared under identical transfer conditions (Supplementary Table 5). In addition to optical imaging, magnified confocal microscopy was used to directly visualize the device–substrate interface and assess local detachment and deformation.

Interfacial integrity was quantitatively evaluated using two complementary error metrics: detached area (%) and geometric strain (%). Detached area represents the fraction of the device that failed to adhere to the mushroom substrate. Geometric strain was estimated from deformation of serpentine interconnects by comparing magnified optical images with the original device geometry, following established geometry–strain correlations (Supplementary Fig. 7).

Elastomer-based carriers (stamps, pads, stretchable substrates): Elastomer-based transfer carriers require full-area, uniform contact to achieve stable adhesion. However, on highly compliant and undercut substrates such as mushroom tissue, the applied pressure primarily

compresses the substrate rather than enabling conformal contact. Visualization using an ultrathin PDMS film confirms the absence of full-area contact, while increased pressure leads to mechanical damage and fracture of the substrate. These limitations are inherent to elastomer-supported stretchable carriers due to unavoidable backing-layer constraints.

Water-based carriers (hydrotransfer printing): Conventional hydro-dipping suppresses pattern spreading using a carrier film, but this configuration is incompatible with highly curved and undercut substrates that require substrate-free electronics. Without a carrier film, uncontrolled lateral water flow induces pattern spreading. During insertion onto curved substrates, fluid motion along the curvature generates strain, misalignment, and loss of pattern fidelity, as clearly observed in optical and confocal images.

High-viscosity liquid-based carriers (molten-sugar analogue): As illustrated in Fig. 2, high-viscosity liquid-based carriers rely on contact through a viscous liquid phase. Because direct molten sugar transfer is impractical, a silica-particle-based high-viscosity fluid was used as an analogue. Interfacial imaging shows pronounced stretching of the electronics on the mushroom substrate, arising from deformation of the viscous fluid that transmits shear and compressive stresses to the device.

DAYS-fluid (this work).

In contrast, DAYS-fluid enabled uniform, full-area conformal contact across the entire mushroom surface, including undercut regions. Owing to its viscosity-controlled, stress-responsive flow, DAYS-fluid adapted locally to complex topographies without compressing the substrate or transmitting shear stress to the device.

Supplementary Table 5 | Comparative feasibility and interfacial integrity of transfer-printing carriers on highly compliant, undercut substrates. Photographic and confocal microscopy–based evaluation of representative transfer-printing carriers, including elastomer-based carriers (stamps, pads, stretchable substrates), water-based hydrotransfer, high-viscosity liquid carriers (molten-sugar analogue), and the DAYS-fluid developed in this work.

		Elastomer (ex. Stamp, Pad, Stretchable substrate)	Water	High viscosity liquid (ex molten sugar)	This work
Optical image							Not Feasible	Not feasible	Not feasible	Feasible
Confocal Microscopy					
		Not Feasible	Not feasible	Not feasible	Feasible
Mechanism		Require full-area contact for adhesion Substrate compression causes non-uniform contact or damage	Uncontrolled fluid flow during floating or dipping	Viscous flow transmits shear and compressive stress	Detailed rheology control enables conformal contact
Quantified interfacial error	Detached area (%)	-	4.75%	8.14%	0%
	Geometric Strain (%)	-	2.08%	5.27%	0%

Supplementary Fig. 7 | Quantification of geometric strain as an interfacial error metric. Confocal microscopy images showing representative electrode patterns transferred using a high-viscosity fluid (left) and DAYS-fluid (right). Geometric strain was quantified by comparing the local curvature and arc geometry of the transferred serpentine traces with the original device design. For each curved segment, the apparent arc angle and radius were extracted from images and referenced to the undeformed geometry, enabling estimation of transfer-induced geometric strain (yellow guides).

[Comment 3] *The demonstration focuses on thermistors. For modalities sensitive to interfacial impedance or strain (e.g., ECG/EMG, ECoG), does DAYS-fluid transfer preserve electrical performance and contact impedance versus standard methods? Benchmarking across modalities would broaden impact.*

[Response 3] We thank the reviewer for this important and constructive comment. We fully agree that demonstrating preservation of electrical performance not only for thermistors but also for sensing modalities sensitive to interfacial strain or impedance would extend the generality and practical relevance of the DAYS-fluid transfer process.

To address this point, we performed additional benchmarking experiments using ultrathin resistive strain sensors, which are inherently highly sensitive to mechanical deformation and interfacial instability. Such strain sensors provide a conservative proxy for evaluating strain- and impedance-sensitive modalities, including ECG, EMG, and ECoG. However, joint regions such as fingers introduce severe folding, which can mechanically damage strain sensors and obscure transfer-induced effects. As a result, to enable a rigorous and controlled comparison, we therefore selected a soft and fragile biological substrate (lettuce leaf) as a model system and directly compared conventional transfer-printing carriers with DAYS-fluid. The corresponding results are presented in Response Fig. 3.

Consistent with the limitations of conventional transfer-printing carriers summarized in Response 2, all existing transfer approaches induced pronounced mechanical and electrical degradation on the highly compliant and fragile lettuce substrate. For elastomer-based carriers, stable transfer requires full-area contact between the electronics and the substrate; however, during contact, the lettuce leaf underwent folding and compression under elastomeric pressure, preventing uniform and conformal adhesion. Consequently, partial contact or complete transfer failure was repeatedly observed (Response Fig. 3c). For high-viscosity liquid-based carriers, direct implementation of molten sugar transfer was experimentally impractical, and a high-viscosity fluid was therefore used as an analogue. As described in Fig. 2, deformation of the viscous carrier during transfer transmitted shear and compressive stresses directly to the electronic device, resulting in visible stretching of the strain sensor and difficulty in maintaining stable electrical performance after transfer (Response Fig. 3d and e). In the case of water-based hydrotransfer, uncontrolled fluid dynamics dominated the process. When the electronics floated on the water surface or when the substrate was dipped into the water bath, hydrodynamic forces pressed and displaced the device, inducing strain and distortion and leading to pronounced electrical instability after transfer (Response Fig. 3f and g).

In contrast, transfer using DAYS-fluid preserved both mechanical integrity and electrical performance on the compliant and fragile lettuce substrate. As shown in Response Fig. 3k–m, the strain sensor transferred using DAYS-fluid exhibited negligible resistance change before

and after transfer and maintained stable, reproducible strain-resolved signals under gentle deformation.

Response Fig. 3 | Comparative evaluation of conventional transfer-printing carriers and DAYS-fluid for strain-sensor integration on soft and hard substrates. a, Calibration curve of the ultrathin strain sensor. **b**, Photograph of a soft, fragile lettuce leaf substrate. **c–g**, Transfer onto lettuce leaf using conventional carriers: elastomer-based carriers cause substrate folding and transfer failure (c); high-viscosity liquid and water-based carriers induce sensor stretching or distortion with corresponding resistance instability (d–g). **h–j**, Strain-sensing behavior on a *soft* biological substrate (lettuce leaf). Optical images of the sensor onto the surface (h). Resistance measured before and after transfer (i). Dynamic resistance traces during low-amplitude deformation of the lettuce leaf. (j).

Beyond epidermal applications, we evaluated the versatility of the DAYS-fluid transfer process using ultrathin strain-sensor electronics (Supplementary Note 8 and Supplementary Fig. 17). On an extremely soft and fragile substrate such as lettuce leaf, DAYS-fluid enabled conformal transfer without tissue damage, preserving baseline resistance and yielding reliable strain-dependent signals. In contrast, conventional transfer carriers, including elastomer-based, water-based, and high-viscosity liquid methods, failed to achieve strain-free integration, causing substrate folding, pattern distortion, or electrical instability (Supplementary Fig. 17c–g). We further extended this demonstration to a hard, textured natural substrate (orange peel), where the strain sensor transferred by DAYS-fluid showed negligible resistance change and stable, reproducible responses under repeated bending, confirming distortion-free integration on rigid, uneven surfaces. – page 18, lines 11-21

Methods

Strain sensor electrical connection and measurement

Electrical connections between the strain sensor terminals and external wires were formed using conductive silver paste (ELCOAT). After application, the silver paste was dried using a handheld hairdryer for 10 min to ensure stable electrical contact. The exposed wiring and contact regions were subsequently encapsulated with a soft siloxane elastomer (Ecoflex, Smooth-On) to improve mechanical robustness, electrical insulation, and user comfort during deformation. Real-time resistance signals of the strain sensors were recorded using a Keithley 2635A source-measure unit and a Keithley 2450 source meter (Keysight Technologies). All electrical measurements were conducted under ambient laboratory conditions.

Supplementary Note 8. Strain-sensor–based evaluation of transfer-induced mechanical and electrical integrity

To evaluate whether the DAYS-fluid transfer process preserves electrical performance for sensing modalities sensitive to interfacial strain and mechanical instability, we performed comparative transfer experiments using ultrathin resistive strain sensors on substrates with markedly different mechanical properties (Supplementary Fig. 17). Strain sensors were

selected as a stringent benchmark because even minor transfer-induced deformation or interfacial instability immediately manifests as measurable electrical artifacts.

The strain sensor exhibits a linear and reproducible relationship between applied bending strain and relative resistance change ($\Delta R/R_0$), confirming its suitability for quantitative evaluation of transfer-induced mechanical effects (Supplementary Fig. 17). We first examined transfer behavior on a highly compliant and fragile biological substrate (lettuce leaf).

Consistent with the limitations of conventional transfer carriers discussed in Supplementary Table 5, existing transfer approaches induced significant mechanical and electrical degradation on the soft lettuce substrate (Supplementary Fig. 17c-g). All conventional transfer methods were performed following the same procedures described in Supplementary Table 5. Elastomer-based carriers failed to achieve conformal contact because applied pressure compressed and folded the substrate, resulting in incomplete adhesion and transfer failure (Supplementary Fig. 17c). High-viscosity liquid-based carriers, employed as an analogue of molten-sugar transfer, transmitted shear and compressive stresses during contact, leading to visible stretching of the strain sensor and pronounced resistance changes after transfer (Supplementary Fig. 17d and e). Water-based hydrotransfer caused uncontrolled fluid flow during floating and substrate dipping, producing pattern distortion, positioning difficulty, and severe electrical instability (Supplementary Fig. 17f and g).

In contrast, DAYS-fluid enabled gentle, deformation-decoupled transfer on the same soft lettuce substrate. The strain sensor transferred using DAYS-fluid exhibited negligible resistance change before and after transfer and maintained stable, reproducible strain-resolved signals during low-amplitude deformation (Supplementary Fig. 17h-j).

To further confirm generality across mechanical regimes, we evaluated strain sensing on a hard, textured natural substrate (orange peel). Sensors transferred using DAYS-fluid showed

unchanged resistance after transfer and stable, repeatable resistance modulation during repeated curvature changes (Supplementary Fig. 17k–m).

Supplementary Fig. 17 | Comparative evaluation of conventional transfer-printing carriers and DAYS-fluid for strain-sensor integration on soft and hard substrates. a, Calibration curve of the ultrathin strain sensor. **b,** Photograph of a soft, fragile lettuce leaf

substrate. **c–g**, Transfer onto lettuce leaf using conventional carriers: elastomer-based carriers cause substrate folding and transfer failure (c); high-viscosity liquid and water-based carriers induce sensor stretching or distortion with corresponding resistance instability (d–g). **h–j**, Strain-sensing behavior on a *soft* biological substrate (lettuce leaf). Optical images of the sensor onto the surface (h). Resistance measured before and after transfer (i). Dynamic resistance traces during low-amplitude deformation of the lettuce leaf. (j). **k–m**, Strain-sensing behavior on a *hard* natural substrate (orange peel). Optical images of the sensor onto the orange peel (k). Comparison of sensor resistance before and after DAYS-fluid transfer (l). Time-resolved resistance signals during repeated curvature changes of the orange peel (m). Source data are provided as a Source Data file.

[Comment 4] *Include process videos. The water-assisted adhesion control and the transfer process are key innovations. Providing short videos of the transfer, adhesion switching, and residue-free removal would greatly improve clarity and reproducibility.*

[Response 4] We thank the reviewer for this helpful suggestion. We fully agree that visualizing the transfer and adhesion-switching processes would enhance clarity and reproducibility. Accordingly, we have prepared short process videos demonstrating the key operational steps of the DAYS-fluid system.

Manuscript

Supplementary Movies 1 to 2

[Comment 5] *Correct minor formatting issues. For example, remove unwanted line breaks on pages 13 and 14.*

[Response 5] We thank the reviewer for noticing the formatting inconsistencies. The unwanted line breaks on pages 13 and 14 have been removed, and the overall layout has been checked for consistency.

Reviewer #1 (Remarks to the Author):

[General comments] The authors have addressed all of my comments, and I am satisfied. I recommend publication of this work in Nature Communications.

[General response] We sincerely thank the reviewer for the careful evaluation of our revised manuscript and for the positive assessment of our work. We greatly appreciate your time and thoughtful review throughout the process. Your constructive comments have helped us improve the clarity and rigor of the manuscript. We are grateful for your recommendation for publication.

Reviewer #2 (Remarks to the Author):

[General comments] I read through the point-by-point rebuttal and the revised manuscript carefully, and I appreciate the amount of work the authors put into addressing the comments. The revision clarifies the motivation and technical positioning much better, and the differences from existing transfer-printing approaches are now laid out more clearly, with additional experiments and analyses to back up the key claims. Overall, the main concerns I raised have been adequately resolved, and I do not have further questions.

I also went through the responses to the other two reviewers' comments, and I feel the authors handled those concerns well too, with thoughtful and convincing revisions supported by new data and clearer explanations. I'm satisfied with the current version and recommend acceptance.

[General response] We sincerely thank the reviewer for the thorough re-evaluation of our revised manuscript and for the generous and encouraging feedback. We truly appreciate the recognition of our efforts to clarify the motivation, technical positioning, and distinctions from existing transfer-printing approaches, as well as the additional experiments and analyses included to support our key claims. We are grateful for your thoughtful assessment of both our responses and the revised manuscript, and we appreciate your recommendation for acceptance.